# Structural Learning in Artificial Neural Networks: A Neural Operator Perspective

**Kaitlin Maile**                                                                *kaitlin.maile@irit.fr*
*IRIT, University of Toulouse*

**Hervé Luga**                                                                   *herve.luga@irit.fr*
*IRIT, University of Toulouse*

**Dennis G. Wilson**                                                  *dennis.wilson@isae-supaero.fr*
*ISAE-SUPAERO, University of Toulouse*

**Reviewed on OpenReview:** *https: // openreview. net/ forum? id= gzhEGhcsnN*

## Abstract

Over the history of Artificial Neural Networks (ANNs), only a minority of algorithms integrate structural changes of the network architecture into the learning process. Modern neuroscience has demonstrated that structural change is an important part of biological learning, with mechanisms such as synaptogenesis and neurogenesis present even in adult brains. Despite this history of artificial methods and biological inspiration, and furthermore the recent resurgence of neural methods in deep learning, relatively few current ANN methods include structural changes in learning compared to those that only adjust synaptic weights during the training process. We aim to draw connections between different approaches of structural learning that have similar abstractions in order to encourage collaboration and development. In this review, we provide a survey on structural learning methods in deep ANNs, including a new neural operator framework from a cellular neuroscience context and perspective, aimed at motivating research on this challenging topic. We then provide an overview of ANN methods which include structural changes within the neural operator framework in the learning process, characterizing each neural operator in detail and drawing connections to their biological counterparts. Finally, we present overarching trends in how these operators are implemented and discuss the open challenges in structural learning in ANNs.

## 1 Introduction

Artificial intelligence algorithms, and specifically artificial neural networks (ANNs), have been able to grow exponentially in power and complexity due to new automated techniques such as backpropagation that replace some of what has been previously hand designed, as well as breakthroughs in computational power. However, many assumptions of current learning approaches such as structurally static learning and isolated training from scratch have diverged from what is observed in the brain, the original inspiration for neural networks. While such assumptions have thus far helped push the frontier of artificial intelligence to its current state by adapting to the characteristics of artificial hardware and software, they may become a bottleneck preventing more powerful ANNs that don't require hand-designed architectures.

Although ANNs with the simplest architecture of an arbitrarily wide single hidden layer are theoretically capable of approximating any function (Cybenko, 1989; Hornik, 1991), deep ANNs with more nuanced structures have taken over in popularity in practice. Computational capacities have improved to allow such large and deep networks. An ANN structured specifically for a given domain can approximate a function more practically and efficiently in storage and training time than a generic wide shallow network (Mhaskar &

Poggio, 2016; Poggio et al., 2017). Even deep dense ANNs, or multilayer perceptrons (MLPs), use structural priors and are not completely dense, as each neuron is only connected to the neurons in the immediately preceding and following layers, rather than all neurons in a network. This imposes a basal amount of structure on the network. Convolutional networks are even sparser than this, limiting neurons to have only spatially local connections and imposing weight-sharing on these connections to create translational equivariance for each layer. Skip-connections can be represented as a special operation within the broader representation of layers connected in series. Although each of these ANN structures are simply special cases of more generic structures, they empirically yield valuable benefits during the learning process. The brain also shows structural specialization with hierarchical organization in task-specific regions, such as the visual cortex (Rash et al., 2016), suggesting the benefit of structure for guiding learning.

Deep ANN architectures are usually hand-designed, static during training, and used generally across many related tasks within a domain. These architectures have begun to approach the desired specialization through their hand-design: however, such deep ANNs are known to be over-parameterized (Ba & Caruana, 2014; Frankle & Carbin, 2018), using more parameters than necessary for their performance on a specific task. Over-parameterization seems to help gradient descent of parameters of an ANN converge to the global optima for a given training dataset, task, and objective function (Zou et al., 2020), particularly given that the architectures are reused for many tasks due to prohibitive engineering costs for further specialization. However, this over-parameterization also comes at a cost of space and time efficiency, during training and deployment. Using even more specialized architectures could reduce this cost and increase the efficacy of the ANN in a specific task.

Automated specialized design of ANN architectures is achievable through structural learning. We define structural learning as optimizing both the architecture and the parameters of that architecture in a single process. As shown in this work, most of this form of structural learning used in practice has taken the forms of pruning, or the removal of parameters to reduce network size while maintaining performance, and Neural Architecture Search (NAS), or the automated search of architectures within a search space. Relatively few papers have pursued developmental structural learning, which involves creation of new structures or connections within the network.

In an effort to surpass the structure design bottleneck, we look to the parallels of ANNs in biological nervous systems in order to define a framework for structural learning in Section 1.1. We survey methods of structural learning in ANNs already existing in literature, providing broad statistical overviews in Section 2 as well as detailed tables of features over our selected corpus at the end of the paper, then diving deeper into each operator of our framework across implementations in Sections 3 and 4, including the biological mechanisms which resemble the patterns of algorithmic components seen in structural learning literature. We finally give suggestions on how to progress forward on current challenges in Section 5.

Our aim in this work is to shed light on a set of similar research directions which are already well established in disjoint fields but not yet often linked. Using a common language for structural learning may help connect the many subcommunities of ANN research that are already researching similar approaches to structural learning with different implementations, abstractions, and goals.

### 1.1 Framework Definitions

In order to understand structural learning, we propose a definition which, in this work, will be used to define the scope of study. We consider structural learning to be a change to a neural network through any of the following four atomic operators, referred to subsequently as the **neural operators**:

- **Neuron creation**: the addition of a unit to an ANN,

- **Neuron removal**: the removal of a unit from an ANN,

- **Synaptogenesis**: the creation of a non-zero weight between two units,

- **Synaptic pruning**: the removal, or change to zero, of a non-zero weight between two units.

We purposefully open the definition of neuron creation and removal to different unit types: in many structural learning algorithms, entire groups or "units" of parameters are added and removed together. Units may be as simple as a neuron in an ANN, or be another higher level structure such as a convolutional filter, channel, layer, or module of layers. A unit contains a portion of the parameters within the network and provides basic organization to make the construction of a network more tractable. While the basic computational units in both the brain and in ANNs are conventionally called neurons, we use unit as a more generic term that can include any structure within a network being used as nodes in a graphical representation of the structure.

The dominant form of learning in ANNs is weight training through error backpropagation, which optimizes synaptic weights and neuron biases in an objective function landscape using gradient descent while keeping the other characteristics of each neuron constant: the layer type, activation function, and pattern of non-zero connections. In this article, we aim to characterize the impact of these neural operators on learning, which permit changes to connectivity and are often coupled with weight training in a bilevel optimization process.

## 2 Characterizing Structural Learning in ANNs

We review relevant implementations of the neural operators in existing ANN literature, focusing on novel algorithms for changing the structure during the course of training an ANN. Using the definitions of the neural operators of neural creation, synaptogenesis, neural pruning, and synaptic pruning allows for characterization of these algorithms from a structural learning perspective, drawing links between different implementations of the neural operators. Some instances of these neural operators are brain-inspired by design, while others are incidentally correlated to biological operations.

In general, ANNs are constructed as a composition of individual components, layers, in a directed and usually acyclic graph. We refer readers to Goodfellow et al. (2016) for a full review of the different basic components of modern ANNs, such as convolutional, pooling, and normalization layers.

### 2.1 Scope of Study

Our 59 collected papers from ANN literature each demonstrate at least one of the four neural operators, neural creation, synaptogenesis, neural pruning, and synaptic pruning, over the course of training an individual ANN via an optimization method. We focus on standard ANNs with parameterized connections, generally with feed-forward, convolutional, or recurrent layers.

ANNs simplify the biological complexity of connection strength between two neurons into a single continuous value. We consider all non-binary changes to this weight, which could represent biological structural change, to be weight training, not structural learning. This includes soft structural methods that do not make any discrete changes to the architecture and thus no changes to the effective dimensionality of the objective function. Only hard structural methods are evaluated, including those which involve a discretization step after soft structural learning, such as Liu et al. (2018).

While synaptogenesis necessarily follows neurogenesis in the brain through different processes, we consider the non-learned creation of connections of a new unit to be encapsulated within neuron creation in ANNs, such as adding a neuron with default connections to all neurons in the previous and subsequent layers. Thus, in our characterization, algorithms are only labeled with both neuron creation and synaptogenesis if there is a connection learning component for existing units in addition to a process for creating new units with a default set of connections.

Many structural search papers enumerate all possible options within their prescribed search space throughout the learning process in order to select the best combination of structures to build the final architecture. For example, some NAS algorithms initialize all possible layers with all possible connections and learn which ones are best. We consider this as entity pruning, rather than entity creation.

A biologically inspired family of algorithms for discovering ANN structures automatically are evolutionary algorithms. These algorithms model biological evolution by maintaining a population and recombining the genes, which may directly or indirectly encode architectures, of successful candidates to produce more candidates. Some such methods meet our criteria for structural learning by persisting network weights

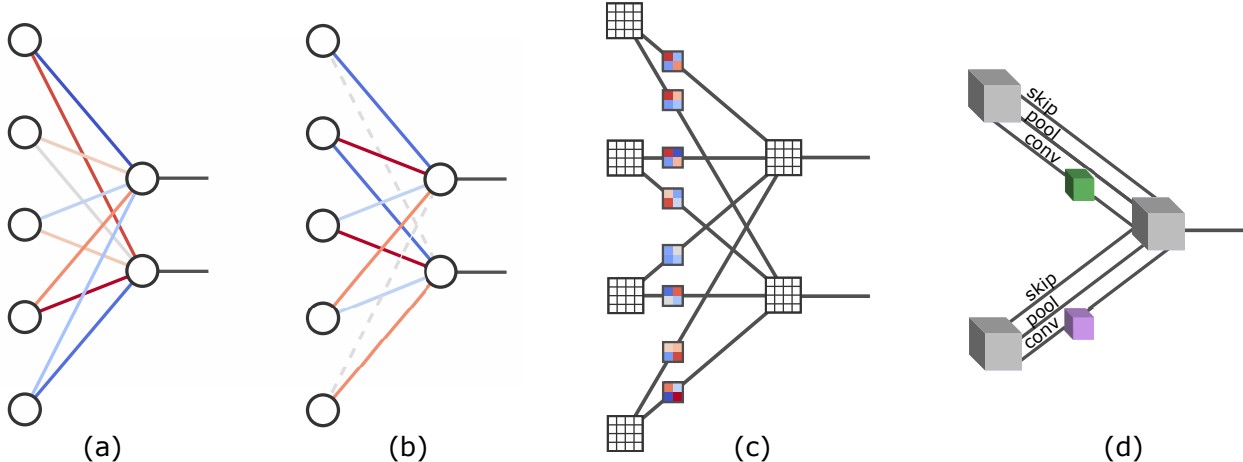

**Figure 1:** Levels of graph abstractions found in structural learning algorithms. From the level of **(a)** feed-forward neurons, there are the two levels of convolutional paradigms: **(b)** on the same level as FFNNs with weight-sharing and systematically pruned connections, and **(c)** on a higher level with a parameter matrix on each synapse and an activation matrix within each neuron, compared to singular values respectively on the lower level. The shared weights in **(b)** are referred to as a kernel, which are the small matrices parameterizing each edge in the higher level paradigm in **(c)**. On an even higher level is the NAS super-network paradigm **(d)**, where each edge may contain parameters based on the operation type, like the parameter-less skip-connection and pooling and the parameterized convolution operations shown. Higher levels have higher dimensionality, structure, and complexity within each node and edge.

across individuals within the population (Elsken et al., 2017; Ci et al., 2021). However, many evolutionary methods maintain a population of individuals with different structures and weights, such as NSGA-Net Lu et al. (2020) and NEAT Stanley & Miikkulainen (2002). These methods include structural learning operators; in NEAT, for example, new neurons or connections can be added through mutation. However, to compare with structural learning in single networks, changes through mutation from one generation to another would have to be studied on specific individuals. We further discuss evolutionary methods in 5, but in the following sections, we only include search methods that which persist weights and study structural changes on individual networks. This limits the collected corpus to one-shot algorithms, which pursue a more continuous path in the changing objective function space during the co-optimization of architecture and parameters.

We exclude architectures with self-gating mechanisms, such as LSTMs (Hochreiter & Schmidhuber, 1997) and transformers (Vaswani et al., 2017), from this framework and corpus. These neural networks change their own structure in a transient and input-dependent manner, occurring on a shorter timescale than structural learning. This is more akin to neural circuitry gating and modulation (Lindsay, 2020). We discuss such dynamic architectures further in 5.2.

The abstraction from individual biological neurons to artificial feed-forward neurons, the most basic type, is clear. However, convolutional neural networks (CNNs) have two levels of abstraction, depicted in Figure 1. The first is by considering a CNN as a special case of a feed-forward neural network (FFNN) with non-local weights zeroed out and local weights shared. The other paradigm moves the neuronal unit up to a higher level, considering a channel in each convolutional layer to be the neuron abstraction. Connections between channels represent synapses and are parameterized by a small matrix kernel rather than a single value, and the activation within the neuron is a larger matrix rather than a single value. While the lower level paradigm continues to multiply the singular activation coming as input to each synapse by the single shared synaptic weight value, the higher level paradigm performs a convolution of the kernel parameter matrix over the incoming activation matrix before summation across connections. Using the lower level paradigm of Figure 1b for structural learning enables partial-area convolution or changing kernel characteristics such as size and striding across neurons and is denoted as "Kernel shape" in Table 1. Using the higher level paradigm of Figure 1c becomes adding and removing channels as neurogenesis and neural pruning, respectively, denoted as "Filter/channel" in Table 1. An even higher level paradigm is super-networks in NAS, shown in Figure 1d,

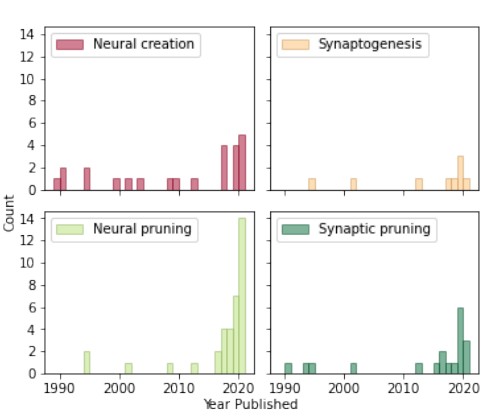

**(a)** Occurrence of operators over time.

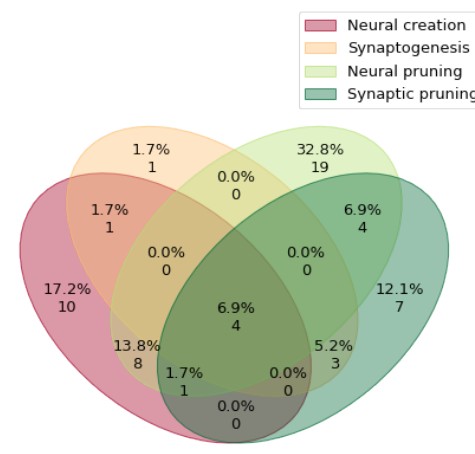

**(b)** Co-occurrence of operators within the collected corpus.

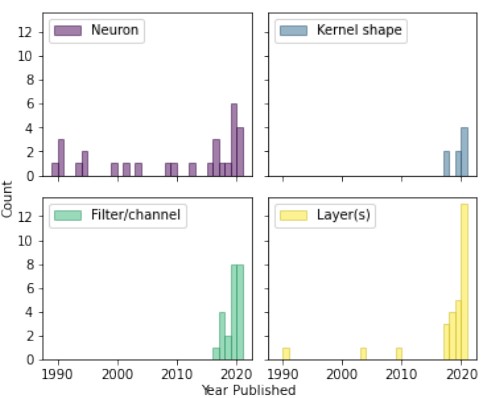

**(c)** Occurrence of unit type over time.

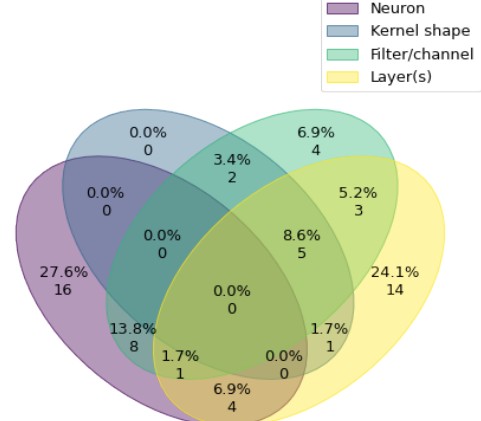

**(d)** Co-occurrence of unit type within the collected corpus.

**Figure 2:** Characterization of neural operators and unit type used in our selected corpus of papers.

denoted as "Layer(s)" in Table 1. We consider works on structural learning which function on these different levels of abstraction.

## 2.2 Characterization of the Collected Corpus

The goal of this work is to characterize structural learning, rather than document all uses of any of the four neural operators in the vast ANN literature. Our selection of 59 seminal works is therefore not comprehensive and is rather intended to be a qualitatively representative sample, as there are orders of magnitude more papers in the subfields of NAS and pruning than in the developmental structural learning space. Completing a more rigorous collection process for complete quantitative characterization would be prohibitively nuanced, for reasons including inconsistent vocabularies surrounding structural learning used across subfields and subtle differences between novel structural algorithms versus applications of existing techniques to new architecture types and task domains, and thus is beyond our scope. The 59 works in the collected corpus provide a diverse and informed view of the structural learning literature.

We refer the reader to other reviews for a more complete view of specific domains. An overview of early works in structural learning and their comparison to biological learning is presented in Quinlan (1998). More recently, Deng et al. (2020) covers pruning and many other methods for model compression. Hoefler et al. (2021) discusses sparsity in neural networks, which is usually achieved via pruning. Elsken et al. (2019)

describes the state of NAS as a burgeoning sub-field, while Xie et al. (2021) is more recent and focuses on weight-sharing NAS. He et al. (2021) covers AutoML, which encompasses methods that contribute to automating the machine learning pipeline beyond parameter tuning and thus includes structural learning as well as data preparation, hyperparameter optimization, and other types of architecture search. Stanley et al. (2019) gives an overview of the use of evolution for neural network optimization, including methods which change neural structure over generations. Evolutionary developmental methods which evolve rules for structural learning are reviewed in Miller (2022). Parisi et al. (2019) discusses methods for continual learning in ANNs, many of which use forms of structural learning to increase the information capacity of networks performing multiple tasks. Han et al. (2021) covers dynamic ANNs, considering adaptability of the network beyond just connectivity during training as in our framework. Our goal in this article is to offer a coherent synthesis of works from these different domains under the framework of structural learning.

The general statistics for neural operators and units across the collected corpus are shown in Figure 2. These plots show trends and frequencies of characteristics across the collected corpus over time, although not necessarily representative of structural learning without the aforementioned biases in our sample collection. Neural creation and synaptic pruning appeared earlier, while synaptogenesis and neural pruning have become more popular in recent years, shown in Figure 2a. Most papers perform a single operator, but a significant portion perform a combination, shown in Figure 2b. Operating on units of standard neurons has been consistently used over time, while the arrival of deep learning architectures is evident in Figure 2c with structural learning over layers and convolutional units of kernel shape and channels becoming more popular after 2010. The impact of deep learning is also evident in Figure 2, with a significant increase in pruning methods while the automatic creation of new units or connections did not undergo a similar expansion of interest. We expect this is partially due to the relative complexity of the creation decision, which we explore in section 4. Finally, as with neural operators, operating on a single unit type is common, but more algorithms are permitting structural learning across different units and layer types, shown in Figure 2d.

Further trends over the qualities of each algorithm in the collected corpus sorted by publication year are detailed in Tables 1-2 at the end of the paper. The earliest algorithms were designed before convolutions and recurrent layers, but now most algorithms are designed for and demonstrated on ANNs with convolutional layers, with some additionally applied to dense layers or recurrent layers. Regarding tasks, nearly all algorithms are demonstrated with supervised learning tasks, particularly image classification in more recent years. Some specific techniques are applied to multi-task, continual learning, more specialized computer vision, natural language processing, sequential data, and reinforcement learning tasks: all of these pose additional challenges above the relatively simple image classification or tabular dataset classification baselines. We discuss existing structural learning methods in the context of each operator in Sections 3-4 and relevant implementation trends in Section 5.1.

Each of the four neural operators have different considerations and effects within structural learning. Performing synaptic pruning or neuron removal on an ANN is more tractable than their additive counterparts, because their structural learning search space is naturally defined by the existing network structure rather than being open-ended. As the two pruning operators have also been more commonly used in literature, we will consider them first in Section 3 before discussing the two creation operators in Section 4.

## 3 Synaptic Pruning and Neural Pruning

Our main questions to characterize pruning in our selected papers are:

- **Goals of Pruning:** Why remove connections or units from an ANN?

- **Biological Inspiration:** How can biological synaptic pruning and programmed neural cell death inspire artificial methods?

- **Architecture Specificity:** How has pruning been adapted to different ANN architectures, such as dense layers and convolutional layers?

- **Measuring Entities:** How is the search performed? How is importance of an entity measured in order to make the pruning selection?

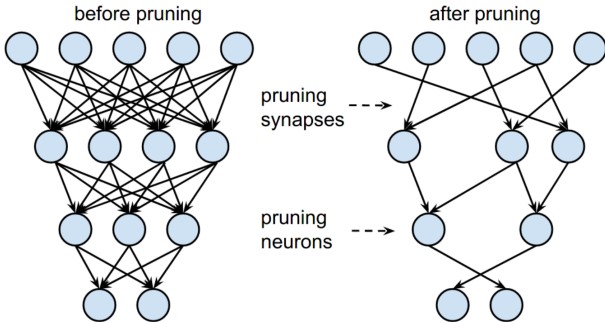

**Figure 3:** Neural pruning and synaptic pruning in a fully connected ANN (figure reproduced with permission from Han et al. (2015)). When a neuron is pruned, as in structured pruning, all incoming and outgoing synapses are also removed. Synaptic pruning additionally removes synapses between unpruned neurons and is also referred to as unstructured pruning.

- **Scheduling Pruning:** When and how often does pruning occur with respect to optimization of the unpruned parameters?

- **Handling Pruned Entities:** How should pruned connections and units be handled?

- **Impact on Weight Optimization:** What is the impact of pruning on optimization of parameter weights?

The following subsections consider the implementations of synaptic pruning and neural pruning found within our corpus. We will first discuss a subset of these questions for pruning in general in Section 3.1 and then dive further into specifics of all questions for synaptic pruning in Section 3.2 then neural pruning in Section 3.3. Finally, we will draw conclusions and connect our discussion back to biology in Section 3.4.

### 3.1 Pruning Generalities

Pruning aims to remove parameters in an optimal manner, but their effect is not independent, and evaluating the effect of all possible combinations of parameters to prune on the network performance is not computationally feasible. Not only do the combinations grow exponentially with the size of the network, but the cost of completely evaluating each one by locating the new optimum through optimization to convergence is prohibitively expensive. Thus, algorithms generally make an approximation by estimating the effect of removing each parameter independently from the local landscape of the objective function on training data (Hoefler et al., 2021). This naturally leads to greedy pruning decisions from the static set of weights, so parameters that currently seem beneficial to remove are selected at the time of pruning. Algorithms can only assume that this process approximates the globally optimal solution for what to remove from a network. Many Neural Architecture Search (NAS) papers formalize the difficulty of optimizing architectures and network weights simultaneously as a bilevel optimization problem, to be discussed further in Section 3.3.

**Goals of Pruning:** Pruning an ANN, or the removal or zeroing of portions of the network, to make it more efficient in space or time while maintaining performance is one of the many orthogonal ways to perform network compression. Using smaller networks with the same performance is particularly desirable for real-time applications and resource-limited devices such as mobile. Pruning large, over-parametrized networks to find their best-performing sub-networks is also a method for finding a high-performing architecture for a given network size. These naturally lead to applications of pruning in multi-task learning, where portions of the network are shared between tasks and others are task-specific, and NAS, where the architecture is optimized on a higher level.

Beyond efficiency, pruning can also yield generalization benefits (Bartoldson et al., 2020; Hoefler et al., 2021). Due to over-parameterization of the base model, pruning can remove learned noise. This is why

many pruning schemes show slightly increased test performance at low to moderate pruning levels. However, the benefit only applies within the initial distribution: pruning may come with a cost of reduced robustness to distribution shifts, but this may be ameliorated by explicit regularization (Liebenwein et al., 2021).

**Biological Inspiration:** Activity-based synaptic pruning is a fundamental component of structuring the information circuitry of the biological brain, integral in varied processes such as memory formation and motor skill learning (Waites et al., 2005; Dayan & Cohen, 2011). Similar to ANN pruning methods such as LeCun et al. (1990); Liu et al. (2021b); Peng et al. (2021), synaptic pruning in the adult brain uses measures of information passage in synapses to remove redundant connections and form sparse functional patterns. Less common is neuron removal, which is most prevalent during early development when neuron removal is the default for new neurons and only those with useful activation patterns survive to maturity (Azevedo et al., 2009). In both cases, the brain relies on the creation of an excess of neural circuitry which is refined into the pertinent functional circuitry through pruning and removal.

**Measuring Entities:** Each entity must be evaluated in order to make an informed pruning selection. The final effect of changing an entity at a given training step cannot be predicted. Entities to prune may be selected by a salience metric computed per-entity designed to estimate the effect of removing the entity on the performance. Masks can also be used, introducing a new parameter for each entity that controls whether it is active or not. This is particularly useful for ephemeral pruning, where pruned entities may be reactivated later in the course of learning. The mask may be discretely controlled by a salience metric or relaxed to continuous values that can be trained via optimization. Finally, regularization can be used to encourage sparsity of network or mask parameters during gradient descent steps by adding a magnitude-penalizing term to the objective function.

Once the selection metric or mask is evaluated, the algorithm may then use either ranking or thresholding, at either a local or global scale, to discretize the pruning decision for each parameter. Ranking allows a defined proportion to be pruned, which may be particularly desirable for hard time or space constraints of the final ANN, while thresholding allows for only parameters meeting a score criteria to be pruned, which can be more consistent over iterative pruning phases. Local pruning makes the ranking or thresholding decision within a structure such as a layer, which makes comparisons more homogeneous, while global pruning makes these comparisons over the entire architecture, which allows the algorithm to determine the pruning level for each structure automatically.

**Scheduling Pruning:** The schedule of pruning determines how it is interwoven with weight updates, including when and how often. Iterative pruning, rather than all-at-once, has generally been more effective (Liu et al., 2021b). It allows the network to evaluate entities in a more intermediate state of pruning, lessening the effects of the assumptions of pruning independence between parameters, high order terms ignored in most metrics, and local objective function evaluations. On the other hand, fully pre-training a network before pruning may be expensive but can be completed independently of the pruning, such as using an off-the-shelf pre-trained model. Most pruning algorithms begin with a large, dense network, while some perform dynamic sparse training, where the initial network is sparsely initialized (Bellec et al., 2018; Wortsman et al., 2019; Liu et al., 2021b).

**Handling Pruned Entities:** After pruning, the pruned entities may be either permanently disregarded or allowed to be revived, which we name ephemeral pruning. We distinguish ephemeral pruning from entity creation if the revival is done with the same method as the pruning and thus is bounded to the same search space as pruning.

**Impact on Weight Optimization:** Because the objective of pruning is often to at least maintain performance, this is synonymous to reducing the dimensions of the objective function space as much as possible while maintaining or decreasing the cost of the found minimum upon convergence of the objective function. Convergence to a globally optimal point is not guaranteed for standard gradient descent techniques in structurally static networks, but the local minimum found usually has a low enough cost in practice (Goodfellow et al., 2016). Because pruning potentially changes the cost and location of each optimum, this is necessarily an even more difficult optimization than training a structurally static network. Techniques mentioned so far such as iterative pruning and regularization intuitively help ameliorate negative effects of the discrete

changes in the shape of the objective function while performing gradient descent in parallel to pruning, thus aiding in the search (Hoefler et al., 2021).

Pruning may be categorized by the level of structure used in the pruning process, as shown in Figure 3. Unstructured pruning, which we discuss further in Section 3.2, is the zeroing of any parameters within an ANN and thus is generally considered synaptic pruning for our neural operator definitions. Structured pruning, which we discuss further in Section 3.3, is the removal of entire units and their associated parameters within a network, such as channels, neurons, or layers, so it is synonymous with neural pruning.

## 3.2 Pruning Connections

**Goals of Pruning:** Synaptic pruning is the lowest level and simplest form of pruning; it intends to remove parameters, which each represent a connection, without any pattern or structure. The earliest pruning papers began with unstructured pruning (LeCun et al., 1990; Hassibi et al., 1993). Pruning synapses in a network is accomplished by zeroing connections and may allow a lower space complexity of storage by storing the remaining parameters in sparse matrices. However it does not yield significant improvements in time complexity of training or inference without specialized software for accelerating sparse matrix multiplications. However, ANNs after unstructured pruning can outperform dense models of the same memory footprint (Zhu & Gupta, 2017), showing that the pruned models can be more specialized and effective, The main applications of synaptic pruning are for network compression (LeCun et al., 1990; Hassibi et al., 1993; Han et al., 2015; Guo et al., 2016; Lee et al., 2019) and for counterbalancing synaptogenesis (Puma-Villanueva et al., 2012; Bellec et al., 2018; Wortsman et al., 2019; Dai et al., 2019b;a; Du et al., 2019). It also benefits multitask learning such as in Peng et al. (2021), where only important connections are saved for the current task and the pruned connections are reused for future tasks.

**Biological Inspiration:** Synapses are continuously being generated and pruned in the adult brain as a part of learning (Dayan & Cohen, 2011). Synaptic pruning in the biological brain can result in weight change, removal of synapses in otherwise connected neurons, or wiring changes, complete disconnection of two neurons (Chklovskii et al., 2004). The latter is the equivalent of ANN pruning, where continous weight values are removed or set to zero. In the brain, this decision is based on activity correlation (Hebb, 1949), where neurons with higher absolute correlation in their activity stay connected while other connections are pruned. This can be modelled as a locally-available measure of importance based on Fisher information (Scholl et al., 2021), similar to unstructured pruning methods in ANNs. However, other decision factors in biological synaptic pruning could be further explored in ANNs. Glial cells have been shown to tag potential synapses for removal using chemical signals over regions of the brain, using spatially and temporally dependent tags (Wilton et al., 2019; Stevens et al., 2007; Riccomagno & Kolodkin, 2015). This allows for coordinated pruning of multiple synapses on different neurons which share characteristics such as a response to certain activity patterns.

**Architecture Specificity:** Unstructured pruning is simple to apply to different neuron and layer types, since it can be agnostic to the layer type and just removes single parameters at a time. For convolutional layers, synaptic pruning can mean removing single values within a 2-D kernel (Han et al., 2015; Guo et al., 2016; Bellec et al., 2018; Dai et al., 2019b;a; Du et al., 2019; Lee et al., 2019) or the entire kernel itself (Wortsman et al., 2019). Although each kernel contains more than a single parameter and is thus a higher structure, it can abstractly represent a connection between channels in convolutional layers, similar to a connection between neurons in dense layers (Wortsman et al., 2019), and cannot be explicitly removed from the 4-D convolutional matrix, only zeroed, so it is more akin to unstructured pruning.

**Measuring Entities:** Salience metrics try to estimate the importance of each connection in order to prune unimportant connections. They can be as simple as the absolute parameter value itself such that weights near zero are pruned (Han et al., 2015; Dai et al., 2019b;a), which is intuitive as they would have a lower impact on the ANN than weights with stronger magnitudes. More complex calculations include gradients, which measure sensitivity of the network outputs with respect to each parameter, so low values signify unimportance. Any metrics requiring a derivative of the objective function must be computed with respect to training data. For most modern datasets, computing them for the entire dataset is not feasible, so mini-batches are often used. A local estimate of the effect of removing a single parameter on the objective function

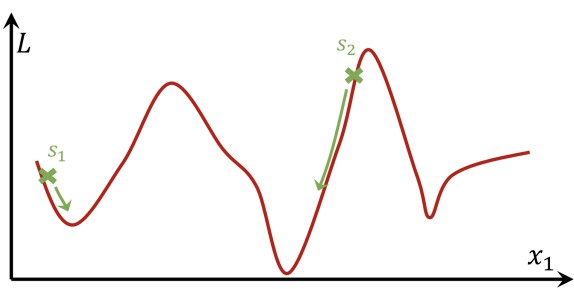
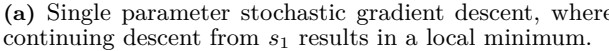
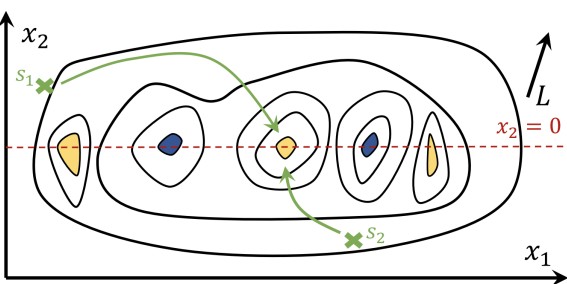

**(a)** Single parameter stochastic gradient descent, where continuing descent from $s_1$ results in a local minimum.

**(b)** Double parameter stochastic gradient descent, where continuing descent from either $s_1$ and $s_2$ results in descending to the global optimum.

**Figure 4:** Comparison of stochastic gradient descent in a toy loss landscape without and with an extra dimension (figure reproduced with permission from Hoefler et al. (2021)). Transitioning from **(a)** to **(b)** represents creation, while transitioning from **(b)** to **(a)** represents pruning. $s_1$ and $s_2$ are potential parameter value(s) at the time of structural change.

may be derived using a Taylor expansion (Laurent et al., 2020). The earliest pruning methods often used up to the second order term, including a full Hessian matrix, within their saliency calculations (LeCun et al., 1990; Hassibi et al., 1993), while current methods usually use cheaper but less accurate approximations with at most first-order derivatives over a single mini-batch (Molchanov et al., 2017; Du et al., 2019; Lee et al., 2019; Peng et al., 2021). For example, the diagonal Fisher information matrix is a first-order metric, scaling linearly with dimensions in computational complexity, that is often used as an approximation of the Hessian matrix, a second-order metric (Peng et al., 2021). The choice of salience metric is usually a trade-off between efficiency and effectiveness, but exactly how useful the precision of saliency is may be confounded with the greedy, iterative pruning process often used: such metrics can only measure local information without interdependencies, so using more expensive methods may be futile.

Masks are also utilized in synaptic pruning. The mask may either be relaxed to be continuous so it can be trained along with the other network weights through optimization (Guo et al., 2016; Bellec et al., 2018; Wortsman et al., 2019; Yan et al., 2019) or remain discretely binary and be controlled with metrics (Dai et al., 2019b;a).

**Scheduling Pruning:** Regarding the weight training portion of the schedule, pre-training the large unpruned network is a common first step to a pruning, but some synaptic pruning algorithms avoid this for training time or space efficiency. Lee et al. (2019) forgoes the pre-training step and instead prunes a newly-initialized model using a first-order derivative metric of connection sensitivity. Hassibi et al. (1993) determines the optimal weight update for the remaining parameters after each pruning, but the inverse Hessian required is too expensive to compute exactly for modern ANNs, although estimations can be used (Hoefler et al., 2021). Most neuron-level synaptic pruning algorithms use fine-tuning optimization steps after pruning to arrive at the final architecture with optimized weights.

**Handling Pruned Entities:** Most metric-based synaptic pruning papers do not allow discrete synapse revival, as computing metrics for inactive parameters often is not possible. However, masked selection techniques are more amenable to ephemeral pruning of connections, which may help ameliorate the negative effects of greediness. Guo et al. (2016); Bellec et al. (2018); Dai et al. (2019a); Wortsman et al. (2019) use dynamic connection search, allowing previously pruned connections to be revived if they show a resurgence in parameter importance later in training.

**Impact on Weight Optimization:** Synaptic pruning may decrease the dimensionality of the objective function search as finely as a single dimension at a time. The proceeding effect conjectured by Hoefler et al. (2021) is demonstrated in Figure 4. This artificial example supports performing more gradient descent in the higher dimensional space and pruning later in the course of training, at least for traditional pruning

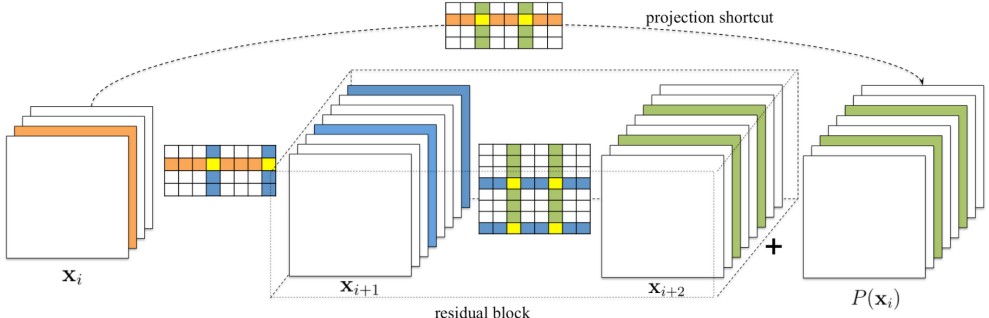

**Figure 5:** Structured pruning in convolutional ANNs (figure reproduced with permission from Li et al. (2017)). Each stack represents the channels of the activation state, while the matrices represent the kernels, which themselves are *nxn* matrices of parameters, for each convolutional layer, with each row being the kernels for each channel in the previous activation and each column being the kernels for each channel in the next activation. Decreasing the incoming channel count in the first convolution is represented in orange. The blue pruning decreases the outgoing channel count of the first convolution and the incoming channel count of the second convolution. This is synonymous with pruning both the incoming and outgoing connections of a neuron in a dense layer. The projection shortcut performs a 1x1 convolution and is pruned accordingly as necessary for dimensionality agreement.

occurring intermittently during training. If the network in Figure 4b is initialized to $s_1$, pruning $x_2$ before the value of $x_1$ surpasses the coordinate of the local maximum may lead to convergence to a non-optimal local minimum, but pruning after this point should allow convergence to the global minimum. Pruning non-zero parameters results in a displacement in the search space based on how large in magnitude the pruned parameters were. Magnitude-based pruning has a minimal but not negligible effect on the displacement of the location in the search space, and may also be more greedy. The performance change due to this displacement does not have clear consequences: the final performance cannot be tractably predicted for any given structural operation. Bartoldson et al. (2020) found that the magnitude of immediate performance drop after an iterative pruning step was positively correlated with improved generalization and performance of the final network, while Laurent et al. (2020) had conflicting findings, showing loosely inverse correlation. While the objective function is an intuitive tool for optimizing the pruning selection, the interacting effects of the many local assumptions and approximations as well as hyperparameters such as algorithmic details confound its use.

### 3.3 Pruning Units

**Goals of Pruning:** Neural pruning is the removal of units within an ANN, such as neurons in fully-connected layers, filters or channels in convolutional layers, or entire layers themselves. Thus, neural pruning has different selection and scheduling considerations compared to synaptic pruning. Structured pruning easily yields both time and space efficiency benefits: entire portions of the network can be removed, thus skipping their computational steps and storage requirements.

Beyond network compression, structured pruning is also a very common abstraction for performing Neural Architecture Search (NAS), an emerging sub-field pursuing the automation of architecture engineering at the layer level. ANN architectures are selected from a predefined search space of possible architectures and evaluated according to a performance estimation strategy. This performance estimation then informs the search, which can be based on evolutionary algorithms, reinforcement learning, gradient descent, or other search methods. Structural learning can be used for NAS by using any of the four neural operators as search operators on persistent networks. Some existing NAS methods function as structural learning, integrating weight tuning with architectural change to arrive at a final static network in terms of both parameters and architecture. Layer-searching NAS algorithms that use structural learning generally employ a hand-designed super-network that contains all possible layer types over all possible interconnections, then perform the operator of neural pruning after optimizing the super-network to derive the final architecture.

**Biological inspiration:** In the adult brain, programmed neural cell death is most common following neurogenesis. During early development, neurons must develop sufficient active connections to avoid the default fate of removal (Yeo & Gautier, 2004). This is similar to one-shot NAS methods such as DARTS (Liu et al., 2018) where large super-networks are created at initialization with only a minority of neurons remaining in the final architecture; however, biological cell removal is progressive and occurs at different rates in different regions of the brain, more similar to Maile et al. (2021). Neural cell death also follows neurogenesis in the adult brain; in the olfactory bulb, only 50% of newly generated neurons survive to be integrated into existing circuitry, with neural activity being a critical factor in determining cell survival (Kelsch et al., 2010). This is most similar to grow-and-prune methods such as NeST (Dai et al., 2019b) and activity-based pruning Rust et al. (1997); Molchanov et al. (2017); Kang & Han (2020), but further research of structural learning methods inspired by adult neurogenesis and subsequent neuron removal is needed.

**Architecture Specificity:** The structural types present within a network require some consideration for structured pruning, depending on the desired level of structure to prune. Many convolutional structural learning algorithms use the higher level paradigm, shown in Figure 1**(c)**, for pruning (Siegel et al., 2016; Li et al., 2017; Du et al., 2019; Wu et al., 2020), as depicted in Figure 5. This allows an algorithm to be applied to convolutional and dense layers, even within the same architecture. Some sub-unit pruning in convolutional layers can still be effective for computational efficiency in contrast to unstructured pruning, such as pruning via striding within kernels (Anwar et al., 2017) and limiting the area of convolution (Dai et al., 2019b). Beyond each layer's type, the surrounding structure and functionality may also be utilized in more specialized pruning: Kang & Han (2020) prunes channels that tend to have low signal after the standard subsequent batch normalization and ReLU operations, while Gordon et al. (2018) regularizes and prunes channels via the batch normalization scaling parameters. Exploiting such known structures in the limited architecture space of those including these structures has a trade-off between search space flexibility and effectiveness within those search spaces. When performing structural learning on units, the algorithm must ensure that all intermediate activations maintain compatibility with all incoming and outgoing layers: for example, neurons tied by skip-connections can be grouped during structural operations Gordon et al. (2018).

**Measuring Entities:** Structured pruning with heuristic search requires metrics or selection methods that can be measured or enforced at the level of desired unit of pruning. Similarly to unstructured pruning, early methods used perturbation-based selection (Narasimha et al., 2008; Puma-Villanueva et al., 2012), where the algorithm measures the performance change from removing each neuron individually and retraining the rest of the network, but this does not scale well with the size of the network. One recent exception used a greedy forward search, where all neurons in a layer are pruned and then useful units are individually added back in (Ye et al., 2020). For metric-based pruning of units, a norm, across parameters within a unit, of any metrics discussed previously for the unstructured case in Section 3.2 can be used (Li et al., 2017; Du et al., 2019; Wu et al., 2020) or group-sparse regularizers can tie parameters within a unit and encourage sparsity at the unit level (Pan et al., 2016; Gordon et al., 2018). Selection methods that are only possible on the unit-level include activation-based pruning, where units with low or infrequent activations are removed (Molchanov et al., 2017; Kang & Han, 2020), and relative metrics of redundancy, where units with redundant functionality compared to others are marked for removal (Siegel et al., 2016). One issue with the latter approach is time complexity, since it requires a pairwise comparison of all units within each layer. Unit-wide masks, where the mask controls whether each unit is active or not, are also common, such as in Chen et al. (2019); Wan et al. (2020); Guo et al. (2021) at the convolutional neuron level and in most NAS works at the layer level.

The two main mask-based developmental NAS approaches at the layer level are continuous NAS and path-sampling NAS, as shown in Figure 6. These are used in order to both train the network parameters and evaluate different architecture possibilities, which may be represented as architecture parameters.

In continuous NAS, a continuously structured super-network is trained and then discretized into the desired architecture, as shown in Figure 6a (Liu et al., 2018; Laube & Zell, 2019; Li et al., 2019; Mei et al., 2020; Yan et al., 2019; Bi et al., 2020; Noy et al., 2020; Wang et al., 2021; Roberts et al., 2021). The structural learning, namely neural pruning of the units of layers, occurs at discretization, which may happen once at the end of the search phase or progressively throughout. Before the super-network is fully discretized, it evaluates mixtures of potential layers over potential connections as weighted sums at each activation state, and uses

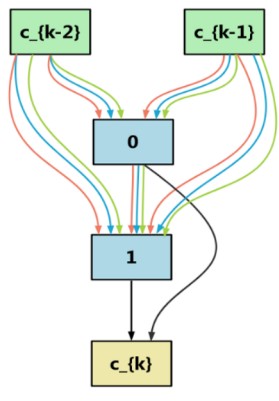

**(a)** Continuous NAS.

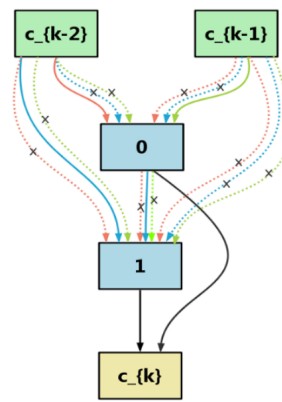

**(b)** Path-sampling NAS.

**Figure 6:** Example of a repeated cell of a super-network in NAS (figure reproduced with permission from Yao et al. (2020)). Within each cell, the blue nodes represent intermediate activations between layers. Each colored edge represents a different layer type, such as a skip-connection or convolutional layer. This cell, $c_k$, receives the output of the two previous cells, $c_{k-2}$ and $c_{k-1}$. During network training, in continuous NAS, all layer options are used at all connections, and a weighted sum is computed at each node using the architecture parameters. In path-sampling NAS, a path representing a valid architecture is selected within the super-network, which is a form of ephemeral discretization. A common architecture constraint for cell-based NAS is at most one layer option per connection and exactly two inputs to each node. For both types of NAS, the final goal is to optimally prune the network into a discrete architecture within this constraint like **(b)**.

back-propagation of the loss error to strengthen or weaken the architectural weight of each layer. Selecting the architecture from the super-network by the end of the search using learned architecture weights often looks very similar to magnitude-based pruning, selecting the strongest options such that a valid architecture is formed.

For path-sampling NAS, potential paths are discretely sampled from the super-network for training the network's parameters within each path, as shown in Figure 6b (Cai et al., 2019; Veniat & Denoyer, 2018; Ci et al., 2021; Guo et al., 2020; Yao et al., 2020; Guo et al., 2021). During the search process, a discrete path is selected for the training step of each mini-batch, and only the network weights and architecture parameters along that path are updated. Thus, each path is equivalent to performing ephemeral neural pruning for a single forward and backward pass, while persisting and updating weights of the super-network. Path selection may use the architectural parameters as sampling or state-transition probabilities within the super-network (Cai et al., 2019; Veniat & Denoyer, 2018; Guo et al., 2020; 2021) or use a uniform distribution (Ci et al., 2021). At the end of the search process, the highest probability path is selected as the final architecture. While path-sampling NAS allows training passes to occur in a discretized network more structurally similar to the final desired architecture than the continuous mixtures in a super-network, only the parameters on the sampled path, rather than all parameters, have gradient information and thus can be updated per mini-batch.

**Scheduling Pruning:** As with synaptic pruning, neural pruning has been trending towards iterative methods, even as often as every iteration (Yuan et al., 2021). In NAS, the earliest continuous NAS methods had a single discretization where the pruning neural operator occurred, after searching the continuous super-network and before evaluating the discretized architecture (Liu et al., 2018). However, this can lead to a discretization gap, where the shallower continuous super-network and deeper discretized architecture are too structurally different and have uncorrelated performance, due to the continuous parameters not being close enough to the discretized parameters (Xie et al., 2021). More NAS algorithms are incorporating progressive discretization (Bi et al., 2020; Wang et al., 2021), among other techniques (Xie et al., 2021), to avoid this gap.

As for the training portion of scheduling, NAS methods also tend to do a full reinitialization and retraining of weights after discretization, whereas most other structural learning methods only fine-tune the network after the search and pruning process, as noted in Table 2. As for network compression, Liu et al. (2019b)

finds that complete reinitialization and training is generally superior while Ye et al. (2020) finds fine-tuning inherited weights is better, provably for the case of shallow networks and empirically for deeper networks.

Avoiding the pretraining step is more difficult in structured pruning than unstructured, since structural pruning selection techniques often assume all units are trained enough in order to differentiate their function from each other and fairly compare their utility. Pretraining is particularly important for comparing parameterized units with unparameterized units in NAS, like comparing a convolutional layer with a skip-connection at the same location within an architecture. Some relatively zero-cost NAS metrics that measure entire architectures without training, such as measuring saliency or sensitivity to perturbation (Abdelfattah et al., 2020) or nonlinearity alignment at initialization (Mellor et al., 2021), have been shown to be effective in identifying good candidate architectures, which is particularly useful to avoid the expensive search training process when the final network is often retrained from scratch.

**Handling Pruned Entities:** At the extreme end of ephemeral pruning on the unit level in frequency is Dropout (Srivastava et al., 2014), which is a technique of randomly and independently selecting units to momentarily prune for each batch during training, preventing co-adaptation and overfitting. Structural pruning algorithms use ephemeral pruning in a more principled and informed manner, as in path-sampling NAS. This allows the modules of the super-network to learn more independently than in the continuous case. This idea is also used by Yuan et al. (2021) at the neuron level, but most other neuron pruning works choose to instead reinitialize new neurons as the creation operation, which will be discussed in Section 4.

**Impact on Weight Optimization:** Structured pruning is less intuitive to consider in the objective function search space, since dimensions are tied through the units containing their corresponding parameters. Thus, multiple dimensions are removed at once for each unit removed. Pruning metrics often gain information through continued training, but this must be balanced with the cost of training units that will be pruned and thus not used in the final network or in further pruning decisions.

Many NAS works pose the architecture search as a bi-level optimization problem, where the architecture parameters $\alpha$ are optimized given that the weight parameters $w$ are optimal for any given architecture and constrained to certain sparsity rules $S$ that only permit valid architectures (Maile et al., 2021):

$$\min_{\alpha} \ L_{val}(w^*(\alpha), \alpha) \tag{1a}$$

$$\textbf{s.t.} \ w^*(\alpha) = \operatorname*{argmin}_{w} L_{train}(w, \alpha), \ \alpha \in S. \tag{1b}$$

This problem is intractable to solve directly, so the process is usually approximated by continuous relaxation of $\alpha$, persisting weights across the architecture optimization search, and alternating weight updates with architecture updates. The gradient updates to the architectures variables may only be locally approximated: using a second-order estimate of the gradient of $\alpha$ that incorporates an inner update of $w^*$ only gives minor improvements in performance for a higher computational cost compared to first order estimates that approximates $w^*$ as the current $w$ (Liu et al., 2018). See Liu et al. (2021a) for a further discussion on bi-level optimization in ANNs.

### 3.4 Pruning Conclusions

In ANNs and in the brain, more information is available for existing neurons and synapses to be pruned versus creating new units. This trend is noted in artificial algorithms, both in the popularity of pruning versus creation as well as more specifically using masking to propagate gradient information to inactive units, thus turning a structural learning problem into a pruning problem.

Pruning in both the brain and ANNs can improve performance relative to cost: the main costs in both medias are time and space, but the realization of these costs change how efficiency is achieved. For example, an important difference between brains and ANNs is the locality of physical biological signals versus the globality of ANN addressing in memory. Biological neurons are limited by physically transported molecular signals, but do not have a scaling time cost for enumerating the local options. ANNs algorithms have no distance-based cost and thus may make decisions globally, but do have a linearly scaling time cost for each entity that is evaluate due to the hardware's bound on floating point operations per second.

Pruning, however, does come with costs, notably the wasted training time of the associated parameters and potential disruption to the learning process. Pruning algorithms must thus strike a balance of information that could be useful for pruning and later utility in the network if the unit is not pruned versus the cost of obtaining this information.

Pruning has been used throughout the history of deep ANNs to partially automate the structural design. However, it relies upon a predefined and instantiated search space. Currently, most methods explicitly hand-design this search space as the initial architecture. Others, as well as the brain, use creation to dynamically create the search space. These creation methods and their synergies with pruning will be discussed in the following section.

## 4 Neuron creation and synaptogenesis

Adding to a search space is naturally more difficult than removing, because an indefinite number of ways to add new elements exists, but removal is limited to only existing elements. For the unit operators of structural learning, this means that neural creation is a much different problem from neural pruning. For the connection operators, because synaptogenesis occurs between existing structures, it is of the same search complexity as synaptic pruning, although metrics for where to add connections are not as straight-forward as where to remove them. Thus, synaptic pruning and synaptogenesis often look very similar, especially for ephemeral structural changes during training.

Nearly parallel to our pruning questions, our main questions for characterizing creation are:

- **Goals of Creation:** Why add connections or units to an ANN?

- **Biological Inspiration:** How can biological neurogenesis and synaptogenesis inspire artificial methods?

- **Architecture Specificity:** How has creation been adapted to different ANN architectures, such as dense layers and convolutional layers?

- **Measuring Entities:** How are the new entities to be added selected over other options?

- **Scheduling Creation:** When and how often does creation occur with respect to optimization of the existing parameters?

- **Operator Interactions:** What is the impact of using pruning along with creation, versus just one such modality?

- **Impact on Weight Optimization:** What is the impact of creation on optimization of parameter weights?

In discussing these questions, we focus on the growth aspects of the papers in our corpus: neural creation and synaptogenesis. We discuss the few topics that span both modalities of creation in Section 4.1, then cover each question specifically for synaptogenesis in Section 4.2 and for neural creation in Section 4.3 before concluding in Section 4.4.

### 4.1 Creation Generalities

Not as many shared characteristics exist between the modes of creation compared to that between the modes of pruning. We detail these few general characteristics below.

**Goals of Creation:** Growing in an ANN allows the architecture to be even more automatically customized: pruning-only algorithms require all entities in the search space to already exist in the initial network. Using the growing neural operators allow a much larger architectural search space to be explored over the course of structural learning, since this search space does not have to be explicitly predefined or instantiated. It is rather implicit from the algorithm's creation operations.

**Biological Inspiration:** Synaptogenesis is a frequent event in the biological brain which defines the possible neural circuitry using information from genetic cues and neural activity (Waites et al., 2005). Neurogenesis, on the other hand, mostly occurs during early development, creating the critical structures of the brain largely through genetic influence (Azevedo et al., 2009; Ackerman, 1992; Urbán & Guillemot, 2014). Neural activity regulates both processes but is especially important for synaptogenesis (Rash et al., 2016; Yoshihara et al., 2009); physical proximity and chemical neuromodulatory signals also influence these processes. As in ANNs, creation is used to expand the possible neural circuitry with relatively uniformed propositions; in both cases, creation is a more difficult problem than removal with less information available.

**Impact on Weight Optimization:** Creating entities in an ANN expands the dimensionality of the parameter search space. Network morphisms are methods of adding entities to an ANN with a specific weight initialization that preserves the functionality of the network (Wei et al., 2016). After using a network morphism to grow units or connections, the expanded ANN may then begin gradient descent at an equivalent location of the previous objective function space but now with expanded dimensions. This guarantees that the new optimal point will have at most the same cost if not lower, since the previous optimum will necessarily be included in the new search space within the subspace where all new parameters are nullified. However, adding too many dimensions increases the time and space costs for the ANN, including in search time if the search space becomes unnecessarily complex. Similarly as for pruning as discussed in section 3.1, creation is generally composed of greedy search decisions in a local search, which can only find local optima.

In the following subsection on synaptogenesis, we focus on algorithms that have a specific method for growing connections, rather than just reviving previously pruned connections, previously discussed as ephemeral pruning in Section 3.2, or creating units with their associated connections, to be discussed in Section 4.3.

## 4.2 Growing Connections

**Goals of Creation:** Synaptogenesis can be used in an ANN to create new connections between existing units. It has been used with pruning for discovering more effective flexible wirings between channels or neurons within ANNs, particularly without an initial overparameterized dense architecture (Puma-Villanueva et al., 2012; Bellec et al., 2018; Wortsman et al., 2019; Dai et al., 2019b;a). In other applications, Kim et al. (2021) augments an ANN's performance after training by adding neuron-level connections directly from hidden neurons to the output, while Elsken et al. (2017) searches for layer-level skip connections, among other network morphisms, concurrent with continual training during evolutionary architecture search.

**Biological Inspiration:** Biological synaptogenesis is an integral part of learning which has been demonstrated in memory and control tasks (Holtmaat & Svoboda, 2009; Dayan & Cohen, 2011). While a considerable part of biological synaptogenesis only reinforces the existing connection between neurons through adding new synapses, previously unconnected neurons can also be linked through synaptogenesis. The information used for this process is based on activity and chemical signatures (Yoshihara et al., 2009); nearby neurons will exchange signaling molecules to make them more receptive to synaptogenesis (Erskine & Herrera, 2007). Physical proximity is an important factor in biological synaptogenesis; for example, in a cortical column, two neurons can connect through a short extension, where connection to a neuron to a more distant region requires investing space and energy in growing axon branches with a more daunting problem of connection choice Chklovskii et al. (2004). There are zones of higher connection density, such as cortical columns which have $10^9$ synapses for $10^5$ neurons, as opposed to the overall higher sparsity of the brain, $10^{15}$ synapses for $10^{11}$ neurons. This sparsity means that, through rewiring, a neuron can drastically change its role in information passage Chklovskii et al. (2004), and synaptogenesis is the main catalyst of rewiring change. Further work similar to Wortsman et al. (2019); Dai et al. (2019b;a) which connect previously unconnected neurons during learning should be explored, as the brain demonstrates the importance of forming new connections during learning. Neural activity could form the basis for possible connection, as in the brain (Faust et al., 2021), by detecting neurons or layers in ANNs which have correlated activity and could be directly linked.

**Architecture Specificity:** Neuron-level synaptogenesis algorithms can generally be applied in both convolutional and standard layers. For convolutional layers, synaptogenesis is generally done on the kernel level (Dai et al., 2019b;a; Wortsman et al., 2019). Only Bellec et al. (2018) performs synaptogenesis on individual parameters within each kernel, but does so randomly on masked connections to balance informed synaptic

pruning. Wortsman et al. (2019) introduces a rewiring algorithm that is agnostic to whether connections are convolutional or dense and removes the constraint of layer organization that engineered deep architectures follow.

**Measuring Entities:** Metrics for synaptogenesis are less straightforward than for synaptic pruning, since the potential connections do not have active information flow. Thus, a common approach is to use a mask that inactivates connections during the forward pass but allows propagation of the gradient to them during the backward pass, providing many of the same metrics as are normally available (Dai et al., 2019b;a; Wortsman et al., 2019). To shrink the quadratic search space of all possible connections, Dai et al. (2019b;a) only allow connections between neurons in neighboring layers, while Wortsman et al. (2019) allows connections across multiple layers but still limited within modules of similar depth. Bellec et al. (2018) also uses a mask, but rather performs random synaptogenesis of a currently dormant connection each time another connection is pruned in order to maintain the sparsity level of the network. While masks bound the search space, that is not necessarily limiting to synaptogenesis since it already can only occur between existing units, whether or not a mask is used.

Beyond mask-based approaches, algorithms often use improved performance as the final metric for adding connections. Puma-Villanueva et al. (2012) ranks all possible connections with a mutual information heuristic amenable to synaptic creation and adds those that improve the performance. Kim et al. (2021) evaluates the effects on performance exhaustively for a smaller search space, only allowing hidden neurons to be connected directly to output neurons. Using improved performance as the selection criteria is intuitive, but is a greedy selection towards the goal of optimizing performance and is expensive to complete exhaustively without heuristics, limited search spaces, or randomized evaluations.

**Scheduling Creation:** Synaptogenesis is usually performed iteratively and greedily. New connections need to be optimized after a random initialization for optimal performance, so synaptogenesis phases are generally followed by weight optimization phases of either only the new parameters or the entire network.

**Operator Interactions:** Most synaptogenesis algorithms also implement synaptic pruning (Puma-Villanueva et al., 2012; Bellec et al., 2018; Dai et al., 2019b;a; Wortsman et al., 2019). Implementing synaptogenesis with synaptic pruning can create dynamically balanced connectivity, which may counteract any negative effects of earlier greedy synaptic additions to escape local optima. The remaining synaptogenesis algorithms only use synaptogenesis out of our two connection operators (Elsken et al., 2017; Kim et al., 2021). This simplifies the search process, but may approach local optima due to greediness.

**Impact on Weight Optimization:** Synaptogenesis adds dimensions to the search space. Because any added synaptic parameters can be set to zero to nullify their function and thus preserve the objective function of the ANN, synaptogenesis can guarantee the new global optimum has at most the same cost. However, depending on the initialization and training algorithms used, the synaptogenesis may displace the current location in the objective function space far enough to descend towards a different, possibly worse or better local optimum. Additionally, adding too many dimensions increases the time and space costs for the ANN, including in search time if the search space becomes unnecessarily complex.

## 4.3 Growing Units

**Goals of Creation:** Neuron creation, or adding units to an ANN, is the most open-ended neural operator. The general aims of neuron creation are to improve performance and to create an architecture that is specifically effective for the desired task without manual design. The resulting networks should have either reduced space and time complexity for deployment at a desired performance level or a better performance potential. Beyond algorithms with these general aims, some NAS approaches incorporate neural creation in the form of changing the architectural search space over the course of training (Laube & Zell, 2019; Ci et al., 2021; Roberts et al., 2021), which provides a larger search space than traditional pruning-only approaches. Both the general and NAS-specific applications allow for a more customized architecture with less hand-design.

**Biological Inspiration:** Neurogenesis in the biological brain is more constrained than in ANNs; neural progenitor cells must be properly located to create new neurons, there must be enough space, and the

associated energy cost is non-negligible. The most common period of neurogenesis, early development, is largely guided by genetic cues (Azevedo et al., 2009), with only highly irregular neural activity patterns disrupting neurogenesis Rash et al. (2016). In humans, adult neurogenesis is constrained to a few regions, notably the hippocampus (Ming & Song, 2011; Ernst et al., 2014), and it is common in other organisms Alunni & Bally-Cuif (2016). Adult neurogenesis could serve as further inspiration for structural learning in ANNs as there are fewer constraints. For example, new neurons in the mature brain first develop input synapses which observe incoming connections before firing action potentials (Kelsch et al., 2010); in other words, new neurons are well-initialized based on surrounding activity, similar to the recent studies of Evci et al. (2022); Maile et al. (2022) in ANNs. The conditions leading to adult neurogenesis are the subject of current research in neuroscience (Christian et al., 2014) and could lead to further inspiration in ANNs.

**Architecture Specificity:** Neuron-level creation usually adds neurons of a predetermined type to existing or new layers. However, since selection techniques are often agnostic to layer type, many neural creation algorithms can be applied to architectures with layers of various types like convolutional or dense (Du et al., 2019; Liu et al., 2019a; Wu et al., 2020; Evci et al., 2022; Maile et al., 2022). Similarly to pruning, growing new filters in a convolutional layer also involves a mapping in the subsequent layer (Du et al., 2019), as shown in Figure 5. Layer-level creation often extends layers with the same type as previous layers (Dong et al., 2020; Wen et al., 2020). Roberts et al. (2021) avoids pre-engineering the layer type selection by parameterizing the layer function in a search space that includes most commonly used layer types in NAS among many others.

**Measuring Entities:** Selecting where and how to add new units to a network is not straight-forward. In order to avoid explicitly bounding the search space, a mask cannot be used as it requires all elements to be instantiated to be associated to an element of the mask. Most approaches limit the instantaneous search space of possibilities, such as iteratively adding a single neuron to an existing layer (Ash, 1989; Fahlman & Lebiere, 1990; Frean, 1990; Lehtokangas, 1999; Ma & Khorasani, 2003; Islam et al., 2009; Dai et al., 2019b; Wu et al., 2020), adding a single new layer beyond the current architecture (Fahlman & Lebiere, 1990; Ma & Khorasani, 2003; Cortes et al., 2017; Wen et al., 2020; Wu et al., 2020), or splitting existing neurons (Du et al., 2019; Liu et al., 2019a; Wu et al., 2020). This results in a simpler problem at each step, but stills gives the algorithm the power of an unbounded search space over the course of training.

The simplest approaches, often combined with some form of pruning, use a generic creation scheme that adds some default number of new units (Narasimha et al., 2008; Gordon et al., 2018). Slightly more informed approaches use the network's objective function as a heuristical measure of success, adding the units that reduce the error the most out of generated options to the network (Cortes et al., 2017; Dai et al., 2019b). Evci et al. (2022) adds neurons that maximize the immediate improvement in performance, while Maile et al. (2022) dynamically adds neurons with unique activations or weights to avoid redundancy. Liang et al. (2018) adds special exponentially activated neurons that provably improve the loss landscape.

**Scheduling Creation:** The objective function is used not only for creation selection but also for iterative scheduling and termination. In early construction algorithms, often a single neuron was added at a time upon convergence of the current network until the desired performance is reached (Ash, 1989; Fahlman & Lebiere, 1990; Frean, 1990; Lehtokangas, 1999; Ma & Khorasani, 2003; Islam et al., 2009; Puma-Villanueva et al., 2012). This approach of a single neuron added between training to convergence does not scale very well with the size of the network or complexity of the task. Many of the recent methods still use performance stagnation during training as a trigger for a neural creation phase, but for adding either many new neurons or an entire layer at a time Elsken et al. (2017); Cortes et al. (2017); Liu et al. (2019a). Otherwise, a manual schedule for iterative phases may be implemented (Narasimha et al., 2008; Du et al., 2019; Guo et al., 2021). Most of these schedules introduce hyperparameters, such as thresholds or durations per each phase of the schedule. These require hand-tuning or optimization for each specific dataset, task, and initial architecture. This is not desirable for generality and automation of algorithms across applications, but selecting hyperparameters is a much smaller search space for manual exploration than that of the architecture being automatically designed. Maile et al. (2022) uses a dynamic schedule that adds neurons while novel directions to explore in either the activations or weights in that layer still exist.

**Operator Interactions:** Most of the recent neural creation algorithms also use neural pruning, while the older ones did not. This dynamic approach not only can ameliorate effects of greedy structural changes, but

also leaves the size of the final ANN as either an objective to optimize in tandem to performance or as an open-ended result. For example, Guo et al. (2021) can adapt initial seed networks to any computational budget, whether smaller, approximately the same, or larger than the original network. Connection operators may also be used in addition to unit operators to make finer adjustments to the network (Puma-Villanueva et al., 2012; Dai et al., 2019b; Du et al., 2019).

**Impact on Weight Optimization:** Neural creation expands the dimensionality of the search space of gradient descent. To determine the starting point in the newly expanded space, the weights of new units need to be initialized before they are trained. Unit-splitting algorithms (Lu et al., 2017; Du et al., 2019; Liu et al., 2019a; Dong et al., 2020; Wu et al., 2020) copy the weights and may add a small perturbation to ensure the units are not redundant if split in parallel. Network morphisms, named by Wei et al. (2016) and used by Islam et al. (2009); Elsken et al. (2017); Laube & Zell (2019); Wen et al. (2020); Evci et al. (2022); Maile et al. (2022), are structural operations that initialize new structures to be null, which generally means initializing new weights to be zero. Most other neural creation algorithms use a similar initialization scheme to the original network, such as random weight initialization. Both unit-splitting without random noise and network morphisms perturb the location of the network in the new space less than methods that use random noise in their initialization. Both cases can be beneficial: staying in a similar location with newly added dimensions allows the path of gradient descent to continue with less disturbance or risk of regressing from the current level of performance, while a small perturbation may help escape local optima or saddle points (Liu et al., 2019a; Wu et al., 2020), as demonstrated in Figure 4.

Liang et al. (2018) theoretically shows that the addition of special exponentially-activated neurons to a basic ANN can make all minima globally optimal. However, no empirical work has shown whether these simple additions affect other aspects of training dynamics and performance. For example, additional saddle points may confound the benefits of no bad local minima (Dauphin et al., 2014).

## 4.4 Creation Conclusions

Creation has been less well-studied than pruning in ANNs, due to the additional complexity imposed by the open-ended search space of creation, which is less cooperative with back-propagation than static architectures and even pruning. Hand-designing neural architectures has also made it less practical and imperative, but the desire for it is continuing to increase alongside computational capabilities.

The brain has rather distinct phases of early development and adult life. These correspond roughly to defining the neural architecture search space, or hand-design of an ANN architecture for the case of conventional non-structural learning, and searching within this space over the process of learning. Most structural learning research focuses on automating and improving the latter process, leaving further optimization of search space design as an open problem. However, the search space design is also synonymously influenced by the genetic priors used in the brain that have been optimized over the course of human evolution, making search space design a much broader problem with a larger scope than one-shot learning. The creation operations allow for a more dynamic approach to search space design.

The brain uses Hebbian learning, or reinforcing effective connection strengths in response to correlations in activity between interacting neurons, as one of many mechanisms for dynamic structural learning (Hebb, 1949). Synaptic scaling, or the post-synaptic homeostatic plasticity (Turrigiano, 1999), and non-synaptic plasticity, such as membrane conductance changes (Mozzachiodi & Byrne, 2010), counterbalance this strengthening and are analogous to weight decay and batch normalization, respectively, in ANNs. The explicit use of Hebbian learning in ANNs, which could be implemented as increasing weight magnitude between artificial neurons with correlated activations, is thus far not as popular as back-propagation but has been done (Miconi et al., 2019). Standard ANNs are not time-dependent, which changes the nature of correlations in activity compared to that of the brain, and ANNs have global information access, permitting back-propagation of errors more easily than in the brain, although there are theories of supervised learning via target activity patterns (Magee & Grienberger, 2020). Dai et al. (2019b) applies this technique to the creation operations and shows that growing connections via larger values of the gradient of the network loss with respect to dormant masked weights is a Hebbian-like rule. This suggests further exploration into appropriate application

for artificial media of Hebbian and other plasticity theories such as behavioral timescale synaptic plasticity (Milstein et al., 2021) and .

As previously discussed in Section 3.4, the physical brain excels at local communication without indexing cost while ANNs can operate globally but with a cost per index. This has implications for effective creation in ANNs: to maintain search efficiency, non-random creation in ANNs, where each potential entity must be indexed and evaluated, should be strategically bounded at a given structural step without overly restricting the size of the overall search space. Creation is not as explicitly bounded in brains: unit creation, including migration, is guided by genetic and developmental factors while synaptogenesis is guided locally by chemical signals, yet there are no restrictions such as having predefined patterns of connections for each neuron. For synapse measurement, the brain does not use simulated heuristics like ANNs often do, but instead selects connections by dynamic processes of repeated synaptogenesis and synaptic removal as directed by various types of plasticity Hebb (1949); Turrigiano (1999); Mozzachiodi & Byrne (2010). Similarly for neurons, the heightened neurogenesis of early development is balanced by the automatic removal of new neurons unless they prove their utility. This suggests an approach for ANNs to make sub-optimal but less expensive additions balanced with more careful pruning to dynamically and efficiently construct an informed architecture.

## 5    Perspectives

With the deepened understanding of how each of the four neural operators work in ANNs, we present our perspectives on the overall state of structural learning. We first note implementational trends that shape recent structural learning works in Section 5.1. We then discuss related domains not completely covered by the proposed framework but which are related to structural learning, particularly those with analogies to the brain, in Section 5.2. We finally discuss current challenges with future directions in Section 5.3.

### 5.1    Implementations and Trends

In order to successfully implement structural learning on modern machines rather than biological media, we note several approaches used commonly across algorithms. The ongoing development of computer hardware and software, which provide the media on which ANNs are implemented, has potentiated each of the innovations in ANNs throughout history. Present computational technology is particularly adept at handling array-based calculations. For example, GPUs drastically speed up ANN training wallclock time particularly with stochastic gradient descent (SGD) and tensor-based architecture parameterizations and data structures. However, they also impose a limit in random-access memory (RAM): ANN training is much more efficient and easier to implement if the entire memory-intensive backward pass of SGD can fit on the GPU's RAM at once. The inference computational cost of an ANN is often measured in the number of Multiply-Adds, or equivalently floating point operations (FLOPs), required per inference of a single input. This value can be used as part of a multi-objective approach, maximizing performance while minimizing the inference cost. The training cost is usually measured in GPU-days, the number of days to train on a single GPU using current hardware and software. Low training costs are a key benefit of structural learning over techniques that instantiate and train a statically structured model for many individual architectures. The following implementation trends are noted across our structural learning corpus, specified in Table 2.

**Modularity:** Dividing a design problem into multiple hierarchical levels and reusing modules learned at lower levels makes it more tractable. This is seen biologically, where symmetry and modularity can be seen from the genetic level, reusing sections of DNA for many proteins that can each have many functions, through the nervous system level, showing bilateral and other symmetries, and even further, as in convergent evolution where unrelated organisms evolve similar structures or functionalities.. For ANN architectures, this entails searching for smaller modular structures that can be used as building blocks to build larger architectures. This technique has already been used in the hand-design of architectures, like ResNet (He et al., 2016). It is also used in structural learning, notably unit creation and pruning where groups of parameters are added or removed as a unit. Modularity has especially been used in NAS in order to reduce the complexity of the search space. Most NAS algorithms that allow flexible interconnectivity use repeated cells of the same architecture during the search process. Some recent NAS approaches incorporate structural learning on multiple levels, such as searching for both layer-level and neuron-level structures (Yan et al.,

2019). NAS algorithms incorporating creation take this even further by also optimizing the layer options used (Laube & Zell, 2019; Ci et al., 2021; Roberts et al., 2021), thus expanding the search space. Modularity is particularly capitalized in algorithms that either duplicate or split units (Lu et al., 2017; Du et al., 2019; Liu et al., 2019a; Dong et al., 2020; Wu et al., 2020). These algorithms reuse neurons, layers, or branches of the network that have already been trained on the present dataset and task as initialization points.

**Masking and Super-networks:** A common approach to allow structural learning is to use masking within the ANN implementation. This mask is an encoding of the structure, enumerating all possible entities and controlling which ones are actively used in a given structure. By relaxing this mask to be continuous, the mask can permit continuous learning where the mask itself is treated like parameters of the network that can be optimized and discretized, such as in pruning or continuous neural architecture search.

When implemented on the layer level for NAS, masking also naturally leads to weight-sharing, where many different architectures are represented with shared parameter data structures but the mask imposes different functionality by specifying the architectural path and thus the parameters used within the super-network. The architectures parameters used for this specification in continuous NAS are usually activated by softmax, gumbel-softmax, or another balancing function in order to enforce scaling across parallel options. Gumbel-softmax and annealing are more recent innovations in activation of NAS parameters to smooth the transition to the final discretization (Guo et al., 2020; Kang & Han, 2020; Noy et al., 2020; Wan et al., 2020). Other NAS papers have introduced non-competitive activation functions (Chu et al., 2020). Once activated, the architecture parameters may then be used for weighted sums of the activations from parallel layer options in continuous architecture search. For path-sampling architecture search, the activated parameters are often used as a sampling probability.

When implemented on the neuron level, masking for synaptic pruning, such as in unstructured pruning algorithms, allows for simpler computations by element-wise multiplication, but does not lead to any computational speed-up without specialized software. Because masks for neural pruning control multiple synaptic parameters per unit with a single masking parameter, they may require regularization to enforce group-wise operations on the parameters during the algorithm, or else the mask may be implemented across dimensions of the parameter matrices if that is compatible with the parameter data structure representation, such as batch normalization scaling factors (Gordon et al., 2018). Like NAS, some neuron-level masking implementations also use temperature in their implementation to smooth the transition from the continuous relaxation to discrete masking (Bellec et al., 2018).

Masking naturally turns architectural selection problems into a pruning problem with a bounded search space. This often creates a limitation in neural architecture search and pruning methods where these methods are used: the maximum size network over the course of search should be able to fit its forward and backward passes on a GPU at the same time, even if the desired final network will be much smaller. For example, super-networks in NAS are often limited by the GPU size: for continuous NAS methods, all options are at least partially activated, resulting in a very large network to evaluate and optimize. In order for the final network discretized from the super-network to also approach the GPU size, many continuous NAS works search in a shallower architecture with fewer channels and then implement the discretized structure in a deeper architecture by repeating cells, which is another benefit of using modular cells. However, this restrains the architecture search to repeatable cells with channels, which is not always applicable for tasks outside of image classification. It also further deepens the discretization gap by contributing to the structural differences between the super-network and the discretized network, as discussed in Section 3.3.

Dynamically sparse training methods avoid the large initial model by sparsely initializing the network and dynamically adjusting which entities are active, allowing the network to maintain a maximum density level (Bellec et al., 2018; Wortsman et al., 2019). This results in a predefined structural search space without limitations on the size of the full search space, but requires structural learning processes that are not memory-bound to fully reap the benefits over standard pruning techniques.

**Interconnectivity:** The brain shows connectivity patterns that are much more flexible and complex than modern standard feed-forward networks, where neurons are organized in layers and each layer is only connected to its immediate neighbors. Neural architectures tended towards this organized structure in the early decades of backpropagation, particularly for the multi-layer perceptron. Recently, the skip-connection broke this simple patterning and improved the state of the art in computer vision and many other tasks (Srivastava

et al., 2015). It allows for much deeper networks to be trained without vanishing gradients and for higher and lower-level features to be used together. Many structural learning algorithms like NAS allow for inter-connectivity flexibility on the layer level, while Bellec et al. (2018); Wortsman et al. (2019) allow even finer interconnectivity outside of layer organization on the neuron level. Including the potential for such flexible interconnectivity in structural learning, rather than using a rigid connectivity backbone, greatly expands the search space and thus adds complexity to the search, but also yields the potential for more powerful networks.

**Weight Reinitialization:** Most neural-level structural learning algorithms are end-to-end, where weights learned in tandem to the architecture are used in the final ANN either without any further training or with only fine-tuning. In contrast, most layer-level continuous NAS approaches require full weight reinitialization and training after architecture discretization. While this does allow the super-network and discretized architecture to have different shapes and training hyperparameters, it is generally more expensive and has a risk of uncorrelated performance between the two stages. This trend is also seen in Frankle & Carbin (2018), although here the pruned network returns to the initial initialization for the remaining weights.

## 5.2 Adjacent Domains

In addition to the external review papers cited in 2.2, we build upon our analysis to mention closely related domains and techniques that are not quite covered by our structural learning framework. They generally cover neural timescales or non-structural operations outside of the scope of our neural operators, but could be used in tandem with or inspire other forms of structural learning.

**Attention and LSTMs:** In contrast to the methods we have considered that specifically use dynamic connectivity and structures during the training process, networks incorporating self-gating mechanisms, such as recurrence and attention, exhibit ephemerality at inference time (Hochreiter & Schmidhuber, 1997; Vaswani et al., 2017). This is most analogous to computation on the neuromodulatory timescale. Attention operates similarly to regulatory firing, instantaneously adjusting computation for a single input without longer term effects using feedback connections (Herstel & Wierenga, 2021). Simpler recurrence structures can also operate on this very short timescale, while LSTMs are more akin to chemical neuromodulation (Marder, 2012).

**Biologically-inspired Networks:** There are a number of artificial neural model and network types which are further inspired by biology. Spiking neural networks (SNNs) use the transmission of discrete spikes, sometimes along multiple synapses between a given pair of neurons, to transmit information (Maass, 1997; Tavanaei et al., 2019), thus allowing for spike-timing dependent plasticity and stochasticity (Kappel et al., 2015). Liquid state machines, a recurrent version of SNNs, have been studied using similar neural operators as the ones studied in this work (Tian et al., 2021). As SNNs imitate the spiking behavior of biological cells, they can be used to model biological networks, including network structure (Kasabov, 2014). Simple organisms such as *C. Elegans* with mapped neural circuits can be modelled with high levels of detail (Olivares et al., 2021); neural circuit policy models, inspired by the brain structure of *C. Elegans*, use cellular dynamics that model biological neurons, including time dependence and sparsity (Lechner et al., 2020). Modelling biological neural structure can help understand how such structures form and their role in learning, and biologically-inspired networks are well-positioned for this.

**Evolutionary Algorithms:** Evolutionary algorithms are a logical choice for NAS due to their flexible problem encoding and high parallelization; in addition, searching for the evolutionary priors for effective learning is a clear inspiration from biology. There is a long history of using evolution to find ANN structure: Miller et al. (1989) was among the first to describe architecture search using a Genetic Algorithm; in Richards et al. (1998), ANNs were evolved to play Go; and NEAT (Stanley & Miikkulainen, 2002) has been applied in domains from image generation (Secretan et al., 2011) to Atari game playing (Hausknecht et al., 2014). Contemporary methods combine evolutionary architectural search with gradient descent on weights (Lu et al., 2020; Ci et al., 2021). In these methods, structural changes usually happen at the generational level, for example as mutations of existing architectures; while this makes them difficult to study in the context of individual neural operators, they can provide insight and comparison for structural learning methods, as highlighted in Stanley et al. (2019).

A different evolutionary domain which takes direct inspiration from the biological evolution of nervous systems is the evolution of structural learning rules, as in Gruau (1994); Miller & Wilson (2017); Miller (2022). In these works, structural learning decisions such as performing neural operators are taken by evolved rule sets, in the form of L-systems (Hornby & Pollack, 2002), grammars (Gruau, 1994), or functional graphs (Miller & Wilson, 2017). In these works, individuals in a population representing different structural learning rules are used to develop ANNs; structural characteristics of the resulting ANNs or the performance of the ANN on tasks are used to inform the fitness and optimization of the various rules. Miller (2022) presents a comprehensive overview of this domain. As demonstrated in this article, structural learning decisions can be difficult to design, with aspects like timing, relation to learning, and information from other parts of the network contributing to the decision. Evolution can aid in this design, and we aim in this article to provide information about structural learning which may help inform these evolutionary approaches.

**Meta-learning:** The fields of AutoML and more specifically meta-learning go beyond optimizing a single model and rather tackle optimization of how a model learns (He et al., 2021; Huisman et al., 2021). Many meta-learning works aim for fast learning, using only a few samples to learn a new task. Humans and other animals developed fast learning, such as easily learning how to toss a new object after learning how to toss other objects, over the course of evolution; we cannot nor need to conscientiously change brain structure to learn a new skill, while genes and structural operations in the brain are separated by many processes that prevent any strong direct effects. Thus, while fast learning itself happens on a timescale between neuromodulation and structural learning, the technique of fast learning was discovered over the course of the much longer timescale of evolution.

Similar hierarchies of automation occur in meta-learning and large language models, even going towards networks building networks (Zoph & Le, 2017; Chen et al., 2022). In these approaches and evolutionary search, there must be hand-design at some level, but engineering at a higher level allows for higher complexity to occur throughout the subsequent lower levels. In the biological case, operations within neural networks are abstracted away from the two controllable variables of consciousness and genes. This suggests continuing setting up artificial systems from progressively higher levels, but also comes with the disadvantage of turning the abstraction between levels into a black box. Similarly, meta-learning works often by training ANNs in the pursuit of learning how to improve learning (Finn et al., 2017; Chen et al., 2022), although the recent trend of massive models with attention mechanisms allowing sparse module activations trained on massive amounts of data and tasks exhibit fast learning as well (Brown et al., 2020).

With the current state of structural learning and adjacent domains in mind, we next present challenges and future directions within structural learning.

### 5.3 Current Challenges and Future Directions for Development

As computational speed and power, as well as datasets, continue to increase, the desire to automate the engineering of ANN architectures is increasing as well. Human engineering is currently a rate-limiting step of ANN architecture innovation. Among discussions already presented throughout this paper on the past and current state of structural learning, we will end our discussion with present challenges and future directions for development.

Many of the benchmark applications that structural learning algorithms are implemented and demonstrated for are well-studied supervised tasks over datasets with static and representative distributions across the training and test partitions. While these provide a consistent benchmark against hand-designed networks and are useful for algorithm development, their utilization is almost paradoxical for one of the powerful potentials of structural learning: automating structural design even for tasks that are not well-studied. Structural learning may allow for more power in less studied and engineered applications, bypassing the need for specific architectural innovations to be designed by hand for every such task. This phenomenon is particularly noted in NAS; automating the engineering of architectures could potentiate the application of ANNs to less-studied tasks, but the majority of NAS papers only report results on the same few image classification datasets for which architectures have already been thoroughly researched, each trial trained independently from scratch, with image-specific search spaces like convolutional layers in repeatable cells. Learning environments that go beyond the basics include multi-task learning (Lu et al., 2017; Dai et al.,

2019a; Guo et al., 2020; Peng et al., 2021), continual learning (Dai et al., 2019a; Li et al., 2019; Wu et al., 2020; Peng et al., 2021; Niu et al., 2021), and reinforcement learning (Hornby & Pollack, 2001; Fu et al., 2020; Miao et al., 2021). These may also approach the dynamic learning environment of the human brain more closely, where representations, labels, and rewards must be internally implemented (Botvinick et al., 2020; Banerjee et al., 2021). Few structural learning papers have implemented their algorithms in these environments thus far, but they represent a deeper level of artificial intelligence.

Our definition of structural learning implies dimensionality changes to the objective function space of ANNs. Structural optimization alongside weight optimization thus prompts a multi-level optimization, with the structure of the network being the outer-most level that defines the space of parameters being optimized on a lower level (Xie et al., 2021). This is explicitly considered in many continuous super-network NAS papers as a bilevel problem (Liu et al., 2018), but the optimization is often simplified to alternating gradient descent steps without theoretical guarantees but with reasonable performance relative to time and computation cost in practice. Multi-level optimization methods are continuously being developed (Huang & Huang, 2021) and may be applied to structural learning, but the trade-off of performance with complexity cost must be considered for effective use.

The level of abstraction, demonstrated in Figure 1, has resulting trade-offs for the structural learning algorithm employing it. Lower level abstractions, such as the direct paradigm of feed-forward neurons, allow for fine-grained tuning of the architecture, but may be too minute for architectures with multiple higher orders of structure and billions of parameters. Higher level paradigms, towards NAS super-networks, impose difficulty by requiring additional structural design in the search space, such as selecting or even structurally optimizing the layer options present in each connection as modules. Combining multiple levels within a single algorithm has thus far been rare (Yan et al., 2019), but presents a powerful direction for structural learning, requiring coordination of each level that have mostly been studied in isolation thus far.

Neural creation is the most difficult form of structural learning because it is open-ended and slow with current methods. However, such a large search space gives it the potential to be the most fruitful in the pursuit of automating architectural design of ANNs. It is represented not only in selecting where and how to add new units, but also in widening the search spaces of potential layer types and connectivity patterns, which are currently rather limited to discrete, hand-engineered choices. In order to surpass this limited space, Bronstein et al. (2021) presents a geometric framework to unite diverse ANN architectures, layer types, and data structure types. Applying these ideas to ANN structural learning could expand the search space by utilizing symmetry-inducing invariances with less restrictions on the input data structure, thus supporting applications even to less naturally tensor-like data structures.

In biology, the genetic code potentiates each neural operator within biological media through transcription then translation to proteins underlying the cellular processes that govern brain structure dynamics. In ANNs, the components of the learning algorithm can be considered like the genetic sequence which determines how ANN learning, both basic weight optimization and the more difficult structural learning, can operate within the computational media of hardware and software given an individual in a learning scenario with training data. The developmental search space is encoded in the algorithm. Most such genetic sequences are hand-designed, but some efforts have been made towards recipe search, where the algorithm itself is automatically designed and optimized. Evolution was the crucial process for discovering the current foundation for biological structural learning, such as discovering useful modular structures and defining the phases of high neurogenesis in early development to dynamic connectivity later in life. Some efforts towards replicating this process are in evolutionary recipe search, where an evolutionary algorithm tries to optimize the entire procedure of structuring and training a neural network. This represents learning on multiple timescales, beyond just the lifetime of a single ANN.

Beyond providing a biologically-inspired framework for structural learning in ANNs, we also intend for this work to strengthen connections between the various sub-communities performing structural learning. We noted a lack of cross-citations both across time and across sub-communities. Many pruning algorithms in the last five years implement the same main abstract methodology as those from three decades ago, but now show drastic improvements in performance (Blalock et al., 2020). While pruning and NAS works are somewhat well-connected within their respective corpora, developmental works using the creation operations

are not, due to both a lack of common vocabulary and a relatively low frequency of new methods. Further, algorithms for more dynamically structured ANNs often compare against their more static counterparts, but not vice versa, and a similar pattern occurs for some of the adjacent domains mentioned in 5.2 versus conventional networks. Our provided framework can help bridge the abstract similarities between algorithms to promote collaboration, not only between the neuroscience and machine learning communities, but also between the sub-communities of the latter.

### 5.4 Conclusion

Structural learning in ANNs is a dynamic and diverse field with a vast potential. Our biologically-inspired framework of neural operators, consisting of neural creation, synaptogenesis, neural pruning, and synaptic pruning, provides a means for synergy not only between neuroscience and machine learning, but also between subcommunities within structural machine learning such as pruning, AutoML, neural architecture search, and developmental neural networks. Furthering structural learning across these approaches could lead to significant breakthroughs beyond the current state of the art in artificial intelligence and machine learning.

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

| Reference | Neural creation | Synaptogenesis | Neural pruning | Synaptic pruning | Neuron | Kernel shape | Filter/channel | Layer(s) | Dense | Conv/pooling | Recurrent | Multi-task | Continual | Image classification | Other computer vision | NLP/sequence | Supervised learning |
|---|---|---|---|---|---|---|---|---|---|---|---|---|---|---|---|---|---|
| | Operator type | | | | Unit type | | | | Layer type | | | Task | | | | | |
| Ash (1989) | + | | | | + | | | | + | | | | | | | | + |
| Fahlman & Lebiere (1990) | + | | | | + | | | + | + | | | | | | | | + |
| Frean (1990) | + | | | | + | | | | + | | | | | | | | + |
| LeCun et al. (1990) | | | | + | + | | | | + | | | | | + | | | + |
| Hassibi et al. (1993) | | | | + | + | | | | + | | | | | | | | + |
| Lehtokangas (1999) | + | | | | + | | | | + | | | | | | | | + |
| Ma & Khorasani (2003) | + | | | | + | | + | | + | | | | | | | | + |
| Narasimha et al. (2008) | + | | + | | + | | | | + | | | | | | | | + |
| Islam et al. (2009) | + | | | | + | | + | | + | | | | | | | | + |
| Puma-Villanueva et al. (2012) | + | + | + | + | + | | | | + | | | | | | | | + |
| Han et al. (2015) | | | | + | + | | | | + | + | | | | + | | | + |
| Guo et al. (2016) | | | | + | + | | + | | + | + | | | | + | | | + |
| Pan et al. (2016) | | | + | + | + | | | | + | | | | | | | | + |
| Siegel et al. (2016) | | | + | | + | | | | + | + | | | | + | | | + |
| Anwar et al. (2017) | | | + | + | | + | + | | + | | | | | + | | | + |
| Cortes et al. (2017) | + | | + | | | | | + | + | | | | | | | | + |
| Elsken et al. (2017) | + | + | | | | + | + | + | + | | | | | + | | | + |
| Li et al. (2017) | | | + | | | | + | | + | | | | | + | | | + |
| Lu et al. (2017) | + | | | | | | | + | + | + | | + | | + | | | + |
| Bellec et al. (2018) | | + | | + | + | | | | + | + | + | | | + | | + | + |
| Gordon et al. (2018) | + | | + | | + | | + | | + | + | | | | + | | | + |
| Liu et al. (2018) | | | + | | | | | + | + | + | | | | + | | + | + |
| Veniat & Denoyer (2018) | | | + | | | | + | + | + | | | | | + | + | | + |
| Cai et al. (2019) | | | + | | | | | + | + | | | | | + | | | + |
| Chen et al. (2019) | | | + | | | | + | + | + | | | | | + | | | + |
| Dai et al. (2019a) | | + | | + | + | | + | | + | + | + | + | + | + | | + | + |
| Dai et al. (2019b) | + | + | + | + | + | | + | | + | | | | | + | | | + |
| Du et al. (2019) | + | | + | + | + | | + | | + | + | | | | + | | | + |
| Laube & Zell (2019) | + | | + | | | + | + | + | + | | | | | + | | | + |
| Lee et al. (2019) | | | | + | + | | + | | + | + | + | | | + | | + | + |
| Li et al. (2019) | | | + | | | | | + | + | + | | | + | + | | | + |
| Liu et al. (2019a) | + | | | | + | | + | | + | + | | | | + | | + | + |
| Wortsman et al. (2019) | | + | | + | | | + | | + | | | | | + | | | + |
| Yan et al. (2019) | | | + | + | + | | | + | + | | | | | + | | | + |
| Bi et al. (2020) | | | + | | | | | + | + | | | | | + | | | + |
| Dong et al. (2020) | + | | | | | | | + | + | | | | | + | | | + |
| Guo et al. (2020) | | | + | + | | | | + | + | + | | + | | + | + | | + |
| Kang & Han (2020) | | | + | | | | + | | + | | | | | | | | + |
| Mei et al. (2020) | | | + | | | + | + | + | + | | | | | + | | | + |
| Noy et al. (2020) | | | + | | | | | + | + | | | | | + | | | + |
| Wan et al. (2020) | | | + | | | + | + | | + | | | | | + | | | + |
| Wen et al. (2020) | + | | | | | | | + | + | | | | | + | | | + |
| Wu et al. (2020) | + | | + | | + | | + | + | + | + | | | + | + | | | + |
| Yao et al. (2020) | | | + | | | | | + | | + | + | | | + | | + | + |
| Ye et al. (2020) | | | + | | + | | + | | + | + | | | | + | | | + |
| Bian et al. (2021) | | | + | | | | | + | + | + | | | | + | | | + |
| Ci et al. (2021) | | | + | | | + | | + | | + | | | | + | | | + |
| Guo et al. (2021) | + | | + | | | | + | + | + | | | | | + | | | + |
| Kim et al. (2021) | | + | | | + | | | | + | + | | | | + | | | + |
| Peng et al. (2021) | | | | + | + | | | | + | + | | + | + | + | + | | + |
| Roberts et al. (2021) | + | | + | | | | + | + | + | + | | | | + | | + | + |
| Sinha & Chen (2021) | | | + | | | | | + | + | | | | | + | | | + |
| Wang et al. (2021) | | | | + | | | | + | + | | | | | + | | | + |
| Yuan et al. (2021) | | | + | | | | | + | + | + | | | | + | + | + | + |
| Evci et al. (2022) | + | | | | + | | + | | + | + | | | | + | | | + |
| Maile et al. (2022) | + | | | | + | | + | | + | + | | | | + | | | + |

**Table 1:** Operator type, unit type, layer type, and task qualities across the collected corpus.

| | Method | | | | | | | | | | Implementation | | | |
|---|---|---|---|---|---|---|---|---|---|---|---|---|---|---|
| Reference | Mask descent | RL generator | Evolutionary | Magnitude-based | 1st order metric | 2nd order metric | Other metric-based | Improvement-based | Regularization | Splitting/joining | Modularity | Weight-share/Mask | Interconnectivity | Final reinitialization |
| Ash (1989) | | | | | | | | + | | | | | | |
| Fahlman & Lebiere (1990) | | | | | | | | + | | | | | | |
| Frean (1990) | | | | | | | | + | | | | | | |
| LeCun et al. (1990) | | | | | | + | | | | | | | | |
| Hassibi et al. (1993) | | | | | | + | | | | | | | | |
| Lehtokangas (1999) | | | | | | | | + | | | | | | |
| Ma & Khorasani (2003) | | | | | | | | + | | | | | | |
| Narasimha et al. (2008) | | | | | | | + | + | | | | | | |
| Islam et al. (2009) | | | | | | | | + | | | | | | |
| Puma-Villanueva et al. (2012) | | | | | | | | + | | | | | + | |
| Han et al. (2015) | | | | + | | | | | + | | | + | | |
| Guo et al. (2016) | | | | + | | | | | | | | + | | |
| Pan et al. (2016) | | | | + | | | | | + | | | | | |
| Siegel et al. (2016) | | | | | | | | | | + | | | | |
| Anwar et al. (2017) | | | + | | | | | | | | | + | | |
| Cortes et al. (2017) | | | + | | | | | | | | | | + | |
| Elsken et al. (2017) | | | + | | | | | | | | | + | + | |
| Li et al. (2017) | | | | + | | | | | | | | | | |
| Lu et al. (2017) | | | | | | | | + | + | | + | | | |
| Bellec et al. (2018) | + | | | + | | | | | + | | | | | |
| Gordon et al. (2018) | | | | | | | | | + | | | | | |
| Liu et al. (2018) | + | | | + | | | | | | | + | + | + | + |
| Veniat & Denoyer (2018) | + | | | | + | | | | + | | | + | + | + |
| Cai et al. (2019) | + | + | | + | | | | | | | + | + | | + |
| Chen et al. (2019) | | | + | | | | | | | | | + | | |
| Dai et al. (2019a) | | | | + | + | | | | | | | | | |
| Dai et al. (2019b) | | | | + | + | | | | | | | | | |
| Du et al. (2019) | | | | + | | | | | + | | | | | |
| Laube & Zell (2019) | + | | | + | | | | | + | | + | + | + | + |
| Lee et al. (2019) | | | | + | | | | | | | | | | |
| Li et al. (2019) | + | | | + | | | | | | | | + | | |
| Liu et al. (2019a) | | | | | | | + | | + | | | | | |
| Wortsman et al. (2019) | + | | | + | | | | | | | | + | + | |
| Yan et al. (2019) | + | | | + | | | | | | | + | + | + | |
| Bi et al. (2020) | + | | | + | | | | | | | + | + | + | + |
| Dong et al. (2020) | | | | | | | + | | | | | | | |
| Guo et al. (2020) | | | | | + | | | | | | | + | | + |
| Kang & Han (2020) | + | | | + | | | | + | | | | + | | |
| Mei et al. (2020) | + | | | | + | | | | | | + | + | | |
| Noy et al. (2020) | + | | | + | | | | | | | + | + | + | + |
| Wan et al. (2020) | + | | | + | | | | | | | | + | | + |
| Wen et al. (2020) | | | | | | | | + | | | | | | |
| Wu et al. (2020) | | | | | + | | | | + | | | | | |
| Yao et al. (2020) | + | | | + | | | | | | | + | + | + | + |
| Ye et al. (2020) | | | | | | | | + | | | | + | | |
| Bian et al. (2021) | + | | | | | | + | + | | | | | + | |
| Ci et al. (2021) | + | | + | + | | | | | | | | + | + | + |
| Guo et al. (2021) | + | | | + | | | | | | | | + | | + |
| Kim et al. (2021) | | | | | | | | + | | | | | + | |
| Peng et al. (2021) | | | | | + | | | | | | | + | | |
| Roberts et al. (2021) | + | | | + | | | | | | | + | + | | |
| Sinha & Chen (2021) | | | + | | | | | | | | | + | + | + |
| Wang et al. (2021) | + | | | | | | | + | | | | + | | + |
| Yuan et al. (2021) | + | | | + | | | | | | | | + | | |
| Evci et al. (2022) | | | | | + | | | | | | | | | |
| Maile et al. (2022) | | | | | + | | + | | | | | | | |

**Table 2:** Method and implementation qualities across the collected corpus.

