# OpenReview forum: "Structural Learning in Artificial Neural Networks: A Neural Operator Perspective"
_TMLR — Accepted by TMLR_

### Review · Reviewer_eh6J · 2022-06-01

**Summary Of Contributions:**

The paper collects 59 DNN papers that apply structural learning (SL), defined (in this paper) as adding or subtracting units or connections from an existing NN during learning. The paper starts with a discussion of biological examples of adding and subtracting units to motivate the DNN work. It then reviews the selected ANN approaches, grouping them according to which operation (add/remove) they apply. This is a survey, so the contribution rests in how material is selected, organized, and summarized.

**Broader Impact Concerns:**

N.A.

**Requested Changes:**

An asterisk (*) means please respond (by clarifying, changing, or other). As to other comments, I think considering them might improve the paper, but it is the authors' choice.

General:

(*) See comment re brain-ANN parallels in "Strengths/Weaknesses".


Abstract:

(*) "biological learning is largely structural": Is this true? I did not see support for this cited in the paper. I mostly got the sense that (i) initial architecture creation is massively structural but occurs without external stimuli (eg in the womb) so it would not correspond to SL as defined in this paper; and then (ii) afterwards learning mostly proceeds via synaptic updates, with small amounts of neurogenesis and rarely apoptosis (if that's the word). An exception would be the massive pruning associated with some stimuli-rich learning phases (eg in human babies).


Page 1:

2 - 3: I don't see how backprop replaces (or is similar to) hand-designed features.

3: big data (eg imageNet) also was critical. This continues to feed ANNs' reliance on big training sets.

8: I would argue that ANNs are not sufficiently hand-designed: they are too general, and rely on training data to arrive at the same result a carefully structured architecture could reach with less training data.

16: a lot of structure, rather than a "small amount"


Page 2:

9: "due to prohibitive engineering costs": and to the research community's strong preference for generalist architectures

10: there is also a high cost in number of training samples needed

(*) 12: is this sentence a fact, or is it a hypothesis? Are there references?

33: "similar set" -> set of similar

38 - 39: "we consider ... the neural operators": very useful definition. It requires as a starting point a "hand-designed" architecture as mentioned above, as well as a hand-designed set of rules for modifications.


page 3:

12 "acutely": meaning?

(*) 28: "early development": The sections on early development seem a bit off-topic, because they involve the initial creation of the neural architecture according to a genetically laid out plan, with internally-generated stimuli (eg retinal waves), but not external stimuli which are being learned. This is different from ANN structural learning, which has exposure to the stimuli to be learned from the start. The mechanism of massive pruning at some phases (eg in babies) is tied to learning, so maybe that could be an inspiration for some ANN approach that starts from over-complete architectures and prunes way down to improve performance on the learned task (not to be confused with pruning to cellphone size while maintaining pre-learned high performance). But I don't see mention of any such approach in the paper (nor do I know of one).

If this observation is correct, then sections 2.1 and 2.2 (and parts of 2.4) could be much reduced.


page 4:

(*) 14: "rarely consider the physical placement of neurons": For ANNs, doesn't "physical placement" mean which layer the neuron is in, and which neurons it's connected to (eg which filter it's a part of)? This is very important to ANNs.

(*) 15 - 18: I don't see how adding new layers or convolutions in an ANN correspond to neurogenesis in a brain.

(*) 19 - 21: "neurogenesis on ... this period": This is not directly related to learning - it is setting up the architecture to be optimized to learn (or just to process information).

(*) 42 - 42: "a number of ... to synaptogenesis": Raises the crucial issue of neuromodulators as mechanisms of weight update. This is widespread in bio brains, mostly ignored in ANNs. How does it fit in to the discussion here?


Page 5:

(*) 4: Are there more references for this? Also, can the Nelson-Alkon ref be moved to connect to this sentence (it's an the exact quote of the first sentence of Nelson-Alkon's abstract).

12 - 14: "From an ... for future learning": It would be helpful to move this statement up to the section that introduces early development (if this section is retained) - the key point is that this category of growth is not relevant to the learning phase in ANNs. It corresponds to the "choose an architecture" step, not the learning step.

20 - 21: "through ... ie pruning": This makes it sound like synaptogenesis is not part of learning - only elimination of synapses affects learning.

(*) 24: "continues throughout lifetime": Is this "pruning" part of learning, or is it part of age/injury-related loss of learning (undesirable), or part of artfully preserving learning to offset shrinking resources?

40: "retinal waves": This is not caused by external stimulus, but is rather internally-generated, non-visual activity.

(*) 42: "deprivation of activity": What does "activity" mean here (and elsewhere): external stimuli, internally-generated things like retinal waves, or both, or something else?


page 6:

(*) 1 - 2: "internal mechanisms of the glial cell tagging": This sounds like a local effect, which contradicts the first clause of the paragraph. Perhaps clarify.

(*) 6 - 7: a) Grammar. b) Is this statement sufficiently supported by the sole example microglia effects? The statement sounds like an ANN analogy for brain region connectivity (mesoscale) effects.

33: typo ("am")

(*) 34 - 38: This paragraph is troubling to me. a) The cell death described above does not apply to entire layers. b) The biological brain may have multiple reasons for large-scale early cell death, which may be specific to the biological context with its many constraints and non-relevant to an ANN context (eg maybe the brain has terrible quality control, and a large percentage of new neurons are defective; this would be irrelevant to ANNs). It is not appropriate to transfer mechanisms from the brain to ANNs just because they exist - there should be some indication that the reason they exist also transfers to the ANN setting.


page 7:

4 - 6: This is a good example of applying the principle described above: just because brains are reluctant to remove mature neurons, it does not follow that ANNs should have the same constraint. Perhaps the costs and benefits of this operation are significantly different in an ANN than in a brain.

(*) section 2.5: I find the connections between bio and ANN tenuous. There are many complexities in brain function that need to be carefully parsed before a neural mechanism can be reasonably said to meaningfully correspond to a mechanism in ANNs. For example, constraints relevant in one domain only (eg the brain's need to be robust to damage, or its need to build on a particular substrate bequeathed to it by evolution; or an ANN's need to run on binary-based digital computers) need to be parsed out to isolate the relevant transferrable mechanism.
The paper could perhaps be strengthened by greatly downplaying the biological angle, which is arguably peripheral, and instead focusing on the well-organized taxonomy of ANN methods.

(*) 42-44: "using ... across neurons": I don't follow this statement. It seems like applying the neural operators at this level results in breaking the shared-weight constraint entirely, because each local convolution gets individually changed by (for example) erasing a connection or input neuron.


Page 8:

8: "all changes to this weight": Does this include synaptic pruning (neural operator d), by setting the weight to approx 0?

(*) 9 - 10: I totally do not get this. Progressive networks add new columns (ie collections of neurons), and also new lateral inter-column connections between neurons, both in direct response to a new learning task. See Rusu Section 2. This appears to be a clear case of neural operators (a) and (c).
Why does the dimensionality of the objective function matter here?

17 - 18: This looks like Rusu 2016 exactly.

19: "all possible options": ...within the prescribed search space

25: "genes": Is this the right word here? If yes, what does it correspond to in the ANN setting?


page 9:

8: This paragraph (ending here) makes good sense (it's of course one of many in this regard - I just happen to call it out).


Page 10:

29: "we expect ... creation decision": Also, it is due to a compelling need to reduce the computational size of DNNs for deployment (especially on phones). This has driven a demand for pruning methods.

36: "demonstrate on": grammar?


page 11:

(*) 26 - 27: "to make it ...while maintaining performance": Is this use-case in the scope of this paper? It is not modification in the service of learning (though it can apply bouts of learning after bouts of pruning, to restore prior performance) - how does this fit into your overall schema?


page 12:

43: typo


page 13:

fig 4b: why are the paths not perpendicular to isobars? (and why does s1 not lead to the local min at left?). Perhaps this is schematic only - however, it is confusing because it contradicts intuition (that gradient descent is perpendicular to isobars).

7: "balancing": What is the purpose - to control overall network size? If yes, would "counteract" or "counterbalance" be better words here?


page 14:

1 - 2: "but exactly ...": This clause is unclear to me.

33: "positively correlated": Does this mean: Larger drops in performance -> larger (ie better) final performance? Or larger negative change in performance -> larger negative change in (ie worse) final performance?


page 15:

15: "batchnorm" vs "batch norm": are both standard usage?

16: "tradeoff of" - "tradeoff between"?

18: missing "the"


page 16:

Caption to fig 6: What are the "0" and "1" blocks?

28: typo


page 17:

26: "many NAS works": If many, why only one reference? Liu 2021a would be good here, since it's a survey.

(*) 41: "locality of phsical biological signals": This appears to contradict the sentence near the end of section 2.3 ("through complex ... over entire regions of the brain")


page 18:

2 - 3: "linearly scaling ...": Can you elaborate on this? Is the cost measured in flops due to summing etc many entities, ie the cost is due to how computers do computation vs how brains do computation?

4: "neural cell removal is costly": Is there a reference for this?

6 - 7: "strike ... this information": Unclear clause.

37: "initial architecture": Don't you always need to hand-design the initial architecture to be grown, as well as hand-specify the rules of allowed growth?


page 19:

(*) 5 - 6: "this guarantees ... if not lower": Can this be claimed for the new Attainable (local) optimal point, in the event that the global optimum is not attainable? ie, the claim may be true for the theoretical global optimum but not for the de facto attainable local optimum. (question)


page 20:

30: "to keep the full search space unbounded": Is this a desirable or possible-to-attain thing? If not, is there a problem with using a mask?


page 21:

22: "network morphisms": what are these?

32 - 33: "neural architecture search space": Is this true, if the brain is not using input stimuli to direct the neural growth?

37: typo "deign" -> design

39: "brain uses hebbian learning":  This sentence gives the impression that Hebbian learning is the only method, which is a strong claim (e.g. what about role of neuromodulators).

39 - 45: Is this paragraph within scope for this paper? It discusses synaptic updates (Hebbian alternatives to backprop), not SL.

50: "instantaneously bounded": meaning?

(*) 50: "creation is not explicitly bounded in brains": This sounds incorrect. All biological systems are significantly bounded in their development options in various ways.


page 22:

(*) 9: The analogy in the 2nd half of this paragraph is unconvincing to me. The [early creation] vs [later tuning based on stimuli] division does not refer to two parts of tuning an existing network, but rather to creating a network and then tuning it via learning. (at least, that's how I understood the descriptions in the early sections of the paper, combined with my own limited understanding)

10: typo

23: typo

25: "one-shot structural learning": What is the definition of "one-shot structural learning"? (I am used to hearing "one-shot learning" refer to learning with one training sample. The overload confuses me.)


page 23:

12: "more native linear algebra": meaning?

21: "should fit ... GPU at once": grammar?

47: "end-to-end": is this standard usage of this term?


page 24:

(*) 4: "As computational ... to increase": What is the role (if any) of the size of training data sets? Are these increasing also, and does this have any implications/effects?

29: "but": is this simplication a bad thing (ie why "but")?


page 25:

4: "genetic code": and circumstances (eg stimuli, epigenetics), and chemical environment, and stochastics

4: typo

(*) 4 - 9: The "genetic code" vs "algorithm" analogy is unconvincing to me - there are too many other factors at work (in brains, genetic code is arguably several steps upstream from the mechanisms actually controlling learning; in computers, hardware is central as well).

21 - 24: This motivation for the paper would also fit well in the Introduction.

**Strengths And Weaknesses:**

The paper collects work from disparate sub-fields of SL (eg NAS, pruning for compression) in one thematic survey. This is a potentially useful perspective/taxonomy, if it connects/introduces readers familiar with one sub-field to other sub-fields in the larger category of SL.


For me, the parallels between brains and ANNs, in terms of SL as defined, are tenuous and not compelling. The paper could perhaps be strengthened by downplaying the biological angle, which is arguably peripheral, and instead focusing on the well-organized taxonomy of ANN methods. For example, the Early Development sections involve the initial creation of the neural architecture according to a genetically laid out plan, with internally-generated stimuli (eg retinal waves), but without external stimuli which are being learned. This is a different scenario from ANN structural learning, which has exposure to the stimuli to be learned from the start.


Weakness of the reviewer: I am not versed in the literature surveyed (beyond the pruning subset), nor in SL. So I don't know if the choice of papers surveyed is appropriate and complete.


Weakness of the TMLR template: Please add line numbers to the template! Line numbers make commenting on a manuscript *much* easier!

---

> ### Author Response · Authors · 2022-06-24
> **Initial response to review (1/2)**
>
> Thank you for your very extensive review. We appreciate your positive and constructive comments; we have added line numbers to the TMLR template to help with the review. In context of the larger changes we made, e.g. restructuring section 2, we respond to all of your specific points for which you specifically requested a response and that are still relevant to the current version below.
>
>
> “(*) "biological learning is largely structural": Is this true? I did not see support for this…”
> We have reworded this statement to “structural change is an important part of biological learning”. Our additions labeled “Biological Inspiration” throughout Sections 3 and 4 support this statement, specifically in the roles of synaptogenesis and synaptic pruning which induce structural rewiring as a part of learning.
>
> Holtmaat, Anthony, and Karel Svoboda. "Experience-dependent structural synaptic plasticity in the mammalian brain." Nature Reviews Neuroscience 10.9 (2009): 647-658.
>
>
> “(*) 12: is this sentence a fact, or is it a hypothesis? Are there references?”: “Automated specialized design of ANN architectures is achievable through structural learning” is meant as an introduction to “structural learning” and is implied by the definition of the term that follows in the next sentence. We posit that the majority of our existing citations, particularly those in our corpus, support the first sentence as they propose structural learning methods which design specialized ANN architectures automatically.
>
> “(*) section 2.5: I find the connections between bio and ANN tenuous…”: As requested, we reduced the focus on the biological angle by removing Section 2 and instead incorporated the most relevant information within the sections on pruning and creation.
>
> “(*) 42-44: "using ... across neurons": I don't follow this statement…”: Yes, it technically does break the shift invariance of convolutions in cases like partial area convolution (Dai et al. 2019b), but not for cases like changing the striding (Wan et al. 2020): in the latter case, neither (b) nor (c) can atomically model striding change as one of the neural operators: (b) requires adding/pruning multiple connections, while (c) changes the number/patterning of parameters within a connection.  This figure is more to show the different levels of units we can think about in the graph representations of ANNs, especially outside of artificial neurons being the units.
>
> “(*) 9 - 10: I totally do not get this. Progressive networks…”: We agree that this example was not beneficial to this section and have removed it.
>
> To clarify our original statement, while progressive networks do perform neural creation, it is triggered not algorithmically but rather when the network begins training on a new task. The progressive network also never goes through a hard discretization or pruning step. We use the dimensionality of the objective function to precisely define when structural learning occurs versus just weight training. Within a single task, progressive networks descend within a static space and do not make a pruning step that could limit the search space, as other works in the studied corpus do.
>
> “(*) 26 - 27: "to make it ...while maintaining performance": Is this use-case in the scope of this paper?...”: Making a network more efficient while maintaining performance is the most common use-case of pruning. Confusion might stem from using multiple meanings of “pruning”, either our two subtractive neural operators, or the more typical sense of using a larger ANN to find a smaller one with at least similar performance. This first sentence of this response refers to the latter meaning, which utilizes our two pruning operators during a learning process to accomplish this goal. That learning process may only involve the pruning operator(s) at a single time, although more successful approaches incorporate pruning operations throughout a parameter training process.
>
> “(*) 41: "locality of physical biological signals": This appears to contradict the sentence near the end of section 2.3 ("through complex ... over entire regions of the brain")”: Even if the brain does manage to send messages across regions, this is still very different from a computer being able to access information at any location of the network for essentially the same cost. Everything in the brain is time and distance dependent, with the media carrying the information affecting exactly how these costs scale.
>
> To be continued below.

---

> ### Author Response · Authors · 2022-06-24
> **Response to initial review (2/2)**
>
> Continued from above.
>
>
> “(*) 5 - 6: "this guarantees ... if not lower": Can this be claimed for the new Attainable (local) optimal point”: Yes: the subspace of the original loss landscape still exists, and the previously attainable optima is still attainable by locking the new entities to be null and descending along the path that made that optima attainable previously. For example, if a new neuron is added with zeroed input weights but non-zero output weights to a network without batch normalization, it will not affect the output of the network and thus is a network morphism (function-preserving), but both the input and output weights will have nonzero gradients. If the zeroed input network weights are not updated along with the rest of the network, the neuron will continue to not affect the output and the network will descend to the same local optima as it would have without the new unit being added (barring differences in random number generation).
>
> “(*) 50: "creation is not explicitly bounded in brains": This sounds incorrect. All biological systems…” We further clarified this sentence: Creation is not as explicitly bounded in brains: unit creation, including migration, is guided by genetic and developmental factors while synaptogenesis is guided locally by chemical signals, yet there are no restrictions such as having predefined patterns of connections for each neuron.
>
> “(*) 9: The analogy in the 2nd half of this paragraph is unconvincing to me. The [early creation] vs [later tuning based on stimuli] division…”: We removed this final sentence and instead added a further example for balancing creation with pruning in the brain earlier in the paragraph.
>
> “(*) 4: "As computational ... to increase": What is the role (if any) of the size of training data sets? …”: You have a valid point that having more data increases the desire for models that can (or need to) harness their potential benefit, although there is a lot of intertwined affects: the large language models don’t work without such large datasets or modern computational speed and capacity, and having these large datasets is driven by having large model that need them. We now also mention data to this statement.
>
> “(*) 4 - 9: The "genetic code" vs "algorithm" analogy is unconvincing to me…”: We have reworded this statement to compare “components of the algorithm” rather than just “algorithm”. You are right that genes are rather abstracted away from structural operations, whereas many current structural learning algorithms are directly interfacing with them. Perhaps this can inspire where structural learning could go, adding more levels of abstraction between the controllable parts being engineered and what happens at a structural level. Our rephrasing of this paragraph goes in this direction.

---

> > ### Comment · Reviewer_eh6J · 2022-07-02
> > **Response to response to review**
> >
> > Thanks for addressing my various concerns, as well as those of other reviewers. In particular, I believe the reorganization, de-emphasizing and distributing the biological content, greatly benefits the paper's impact.
> > One last comment:
> > I believe lines 13, 59 - 64, and also 65 put too strong an emphasis on the bio angle, perhaps because they are from the original version. I would suggest that you recalibrate these lines to clarify that you are bringing in bio perspectives as appropriate, rather than saying that it drives your taxonomy and analysis.
> > Also, parentheses want repair in 332 and 487.
> > I am happy to recommend this paper for publication (subject to other reviewers' concurrence of course). My thanks to the authors for all their time, work, and care.

---

> > > ### Author Response · Authors · 2022-07-04
> > > **Response to latest reponse**
> > >
> > > Thank you for your latest response, especially reviewing our changes to provide additional feedback and your recommendation. We have addressed the parentheses as well as reworded the portions of the abstract and introduction that refer to the biological aspect of our work.

---

### Review · Reviewer_WXVF · 2022-06-06

**Summary Of Contributions:**

In this paper, the authors review of structural learning in neural networks. Structural learning is categorized into four "neural operators" -- synapse addition and deletion, neuron addition and deletion. A brief review of development in the brain is given, and analogies with structure learning in ANNs is are made, following which the authors do an comprehensive review of structural learning in ANNs.

**Requested Changes:**

- Provide a tad bit more detail about pre-training (Sec. 4.2, 4.3) that's used before pruning, since otherwise it's not clear how it fits into the schedule.
- In Sec. 4.3,
  - under architectural specificity, some discussion beyond handling convolutional networks is required.
  - There seems to be almost no mention of activity-based pruning of units (e.g. Rust et al. 1997; Molchanov et al. 2019).
  - "Some relatively zero-cost NAS metrics that measure entire architectures without training have been shown to be effective": Would be useful to have these metrics briefly described
  - "while Ye et al. (2020) finds fine-tuning inherited weights is better for pruning algorithms that can effectively use these weights": Would be useful to have such pruning algorithms briefly mentioned or described if possible.
  - In Handling Pruned Entities: Dropout comes out of nowhere, but should also cover how pruned entities are handled in other mentioned algorithms.
- Section 5.4
  - overemphasizes hebbian plasticity. There are many other forms of plasticity found in the brain e.g. see (Magee & Grienberger 2020) and papers on dendritic plasticity. So this section should be balanced a bit more.
  - "For synapse measurement, the brain does not use simulated heuristics like ANNs often do, but instead selects connections by dynamic processes of repeated synaptogenesis and synaptic removal in tandem with Hebbian learning.": Citation needed
- Section 6.1: (Srivastava et al. 2015) was in fact an earlier paper which proposed skip-connections, and should be cited instead of or in addition to (He et al. 2016)
- Section 6.2: (Blalock et al. 2020) seems to be a misplaced citation.

*References*:

Rust, A.G., Adams, R., George, S., Bolouri, H., 1997. Activity-based Pruning in Developmental Arti cial Neural Networks 10.

Molchanov, P., Mallya, A., Tyree, S., Frosio, I., Kautz, J., 2019. Importance Estimation for Neural Network Pruning. Presented at the Proceedings of the IEEE/CVF Conference on Computer Vision and Pattern Recognition, pp. 11264–11272.

Magee, J.C., Grienberger, C., 2020. Synaptic Plasticity Forms and Functions. Annual Review of Neuroscience 43, null. https://doi.org/10.1146/annurev-neuro-090919-022842

Srivastava, R.K., Greff, K., Schmidhuber, J., 2015. Highway Networks. arXiv:1505.00387 [cs].


**Strengths And Weaknesses:**

Strengths:
- The overall categorisation is very well thought out and useful.
- The sub-structure in each category (goals, measuring entitites, scheduling etc.) also makes the review easy to read.
- The review seems to be quite comprehensive, even if not exhaustive, and really makes a clean summary of each of the structural learning methods discussed.
- Overall the writing is clear and understandable, as is the structure.
- The meta discussion of the papers in the review corpus, and the related plots is very nice.

Weaknesses:
- Evolutionary methods that maintain a population (e.g. Lu et al. 2020; Stanley & Miikkulainen 2002) could be discussed a bit more. The authors do mention them, and I agree it doesn't cleanly fit into the categorization of this paper. But they are important structural learning algorithms (IMO) and worth having a section.
- Section 2 on biology, while very interesting and useful to justify the categorization of structural learning, doesn't give much insight into possible algorithms being used in biology (but probably more a limitation of the state of knowledge in neuroscience).
- Section 5: overemphasizes Hebbian plasticity, while many other forms of plasticity exist in the brain.

---

> ### Author Response · Authors · 2022-06-24
> **Response to initial review**
>
> Thank you for your informative review. We appreciate your feedback and address your constructive comments below.
>
> Evolutionary methods: We extended the paragraph about evolutionary methods in Section 2.1 and added two paragraphs in Section 5.2 on evolutionary methods.
> Biology: We removed the section on biology and instead incorporated the most relevant information and how it relates to our corpus within the following sections on pruning and creation.
> Plasticity: We have expanded this paragraph to mention other types of plasticity and how they are related to structural learning.
> Pre-training: We tweaked some wording to help convey that the full process from initialization to the final architecture are part of the structural learning process, even if there are full phases of training without structural change such as in this pre-training case. We also added some details to pretraining in Section 3.1.
> Pruning units:
> -beyond handling convolutional networks: we added some further detail on maintaining consistency of activation shapes, but we also note that the majority of papers within our framework are focused on dense and/or convolutional layers.
> -activity-based pruning of units (e.g. Rust et al. 1997; Molchanov et al. 2019): Molchanov et al. 2019 only uses the first order Taylor expansion term (weight times gradient for each parameter). Perhaps you were referring to Molchanov et al. 2017? In this paper, activation-based pruning performed worse than the taylor-based approach that they use in further works, but we added this citation in addition to Rust et al. 1997 to relate to brain mechanisms for removing low activity neurons as well as to complement Maile et al. 2022 which uses activation information for neurogenesis. We note that Kang et al. 2020 also does activity-based pruning of channels but was previously only mentioned in “Architecture Specificity”, so we added this more explicitly when we discuss activity-based metrics.
> -zero-cost NAS metrics: We added the examples of “measuring saliency, sensitivity to perturbation, or nonlinearity alignment at initialization”.
> -fine-tuning inherited weights: We reworded the end of this sentence: “...while Ye et al. (2020) finds fine-tuning inherited weights is better, provably for the case of shallow networks and empirically for deeper networks.”
> -Handling Pruned Entities: We expanded this paragraph.
> "For synapse measurement, the brain does not use simulated heuristics like ANNs often do…”: We added the plasticity citations also used elsewhere.
> Srivastava et al. 2015: We updated it to this citation.
> Blalock et al. 2020: We fixed this citation placement.

---

### Review · Reviewer_KVFC · 2022-06-13

**Summary Of Contributions:**

This review paper provides an overview of ANN methods which include structural changes within the neural operator framework in the learning process. The article attempts to connect the different methods of structural learning in artificial neural networks (ANNs), and how these methods compare to similar processes in the brain. The authors argue that many current learning approaches in ANNs diverge from what is observed in the brain and that this divergence may be a bottleneck preventing more powerful ANNs. Additionally, the authors argue that automated specialized design of ANN architectures is achievable through structural learning and that using a common language for structural learning may help connect the many subcommunities of ANN research that are already researching similar approaches to structural learning with different implementations, abstractions, and goals.

**Broader Impact Concerns:**

None.

**Requested Changes:**

I asked for further explanations, provided feedback, and outlined what I believe can turn this work into a great publication. In its current structure, I am leaning towards a rejection. However, once the authors address all 4 points raised above, I vote for acceptance.

**Strengths And Weaknesses:**


### Strengths

+ Relevance: Connecting principles of computation in brains and in deep networks is of great interest to a large group of ML readers.
+ Relevance: Obtaining a unified framework in which we can formulate NAS, AutoML, and pruning is another interesting idea.
+ Idea: The proposed neural operator framework is a great way to relate NAS, AutoML, and pruning concepts.
+ Organization: The paper is very well written and easy to follow.

### Weaknesses

1 ) **Ignoring System 2.**  What is the overarching objective of connecting NAS, AutoML, and pruning with their similar processes in brains? I believe the report does a relatively good job at *describing similarities and related works* but fails to provide specific connections between the ways brains perform neural operators and their objectives that could enhance NAS, AutoML, or pruning. In other words, I see a disconnection between findings from neuroscience and **how** and **why** bringing them to our attention is useful to describe NAS, AutoML, and pruning methods included in the report under the umbrella of structural learning.

I really like how the authors organized the characterization of pruning for example into the Goal of Pruning, Architecture Specificity, Measuring Entities, Scheduling Pruning, Handling Pruned Entities, and Impact on Weight Optimization. However, I do not see how and why the biological principles of computation are relevant and how bringing those insights is useful in this study? One could simply discuss, characterize structural learning, and write the review solely on deep models and methods.

Nevertheless, the discussion of the developmental adaptation of biological networks in nervous systems is far less understood, and their objectives are fundamentally different from those artificial specialist systems and methods described in this work. In this context, the ideal review would inspire how system 1 level structural learning gives rise to system 2 level behavior? This is extremely important as this work is emphasizing the connection of brains and artificial systems in structural learning. Why? Because brains have evolved and trained on a large corpus of evolutionary data while interacting in changing environments; thus, giving rise to further neural adaptation mechanisms. In the case of neural network models, we have seen a very similar pattern on large-scale models that are trained on a large corpus of data and then are used either zero-shot or few-shot on specific tasks.

Unfortunately, the current report entirely discards discussing models based on attention that have shown promise in demonstrating both system 1 and 2 level skills during the last 5 years, therefore, becomes significantly limited in impact.

Great starting points for discussing this recent line of work in the current manuscript would be the following:

[1] Nikhil Singh, Brandon Kates, Jeff Mentch, Anant Kharkar, Madeleine Udell, and Iddo Drori. Privileged zero-shot automl. CoRR, abs/2106.13743, 2021.

[2] Chen, Yutian, et al. "Towards Learning Universal Hyperparameter Optimizers with Transformers." arXiv preprint arXiv:2205.13320 (2022).

2 ) **Missing discussions on generalization.** At the core of any learning and meta-learning approach is their ability to give rise to a system that can generalize well and robustly to unseen data. There have to be discussions regarding the generalization of methods described compared to each other. Moreover, discussions on the robustness of sparser networks, or obtained networks from NAS algorithms have to be discussed. The following reference is a good starting point to include a discussion on the robust generalization of sparse networks:

[3] Liebenwein, Lucas, et al. "Lost in pruning: The effects of pruning neural networks beyond test accuracy." Proceedings of Machine Learning and Systems 3 (2021): 93-138.

The authors also discussed "creation" and pointed out that it is much more challenging to perform because of the vast algorithmic space. In [4], It was shown that adding one neuron can eliminate all bad local minima! It would raise the impact of the current work significantly if the authors could add a section on generalization and investigate how the theoretical results of [4] have been observed maybe empirically in the context of NAS.

[4] Liang, Shiyu, et al. "Adding one neuron can eliminate all bad local minima." Advances in Neural Information Processing Systems 31 (2018).

3 ) **Missed bio-inspired sparse networks** I did not see any related work and discussions on advanced neural network architectures that are directly inspired by biological priors and show great promise in modeling data. An Example includes neural circuit policies (NCPs) [5], which are sparse neural networks inspired by the nervous system of the worm C. elegans. I highly recommend discussing this line of work in the revised manuscript, which lies under the class of liquid neural networks [6]. In particular, how they fit into the characterization of structural learning as models that are directly taken from neuroscience.

[5] Lechner, Mathias, et al. "Neural circuit policies enabling auditable autonomy." Nature Machine Intelligence 2.10 (2020): 642-652.

[6] Hasani, Ramin, et al. "Liquid Time-constant Networks." Proceedings of the AAAI Conference on Artificial Intelligence. Vol. 35. No. 9. 2021.

4 ) **Statements that I think are not accurate or need significant refinement and further explanations, or revision:**

4.1. Authors say: "Finally, we exclude architectures with self-gating mechanisms, such as LSTMs (Hochreiter & Schmidhuber, 1997) and transformers (Vaswani et al., 2017). These neural networks change their own structure, but in a transient and input-dependent manner, occurring on a shorter timescale than structural learning. This is more akin to neural circuitry gating and modulation Lindsay (2020)."

I do not see much advantage of this review paper if Transformers and gated RNNs are being left out. Especially because the paper emphasizes brain inspiration; e.g., evolutionary structural learning in brains, their post-evolution adaptations, and their mechanism of memory. Also, neural circuits in brains are highly recurrent. How can we exclude these architectures?

4.2. Authors say: "In biology, the genetic code potentiates each neural operatorwithin biological media. In ANNs, the algorithm can be considered like the genetic sequence which determine exactly how ANN learning, both basic weight optimization and the more difficult structural learning, occurs within the computational media of hardware and software given an individual in a learning scenario with training data."

Based on the discussions above on Systems 1 and 2, this analogy is very coarse. What is the objective? Are we going to discuss general intelligence? or are we just discussing specialized neural networks? Authors must elaborate much more on the claims they make and the similarities between natural and artificial learning systems.

4.3. At the beginning of Section 6.2 the authors say: "As computational speed and power continues to increase, the desire to automate the engineering of ANN architectures is increasing as well. It will be a crucial step to continue in the direction towards general intelligence."

This statement emphasizes the essence of discussing pre-trained large-scale models built by Transformers. Without discussing those models how can we conclude something about general intelligence?

---

> ### Author Response · Authors · 2022-06-24
> **Response to initial review**
>
>
> Thank you for your comments. Many of the suggested citations are interesting approaches which prompted further discussion in the Perspectives section (now part 5) and which have clear links to structural learning. However, we did not include certain in our defined scope of study of neural operators for structural learning and thus they were not added to our corpus of papers described in the tables and summary statistics. We clarify these decisions below.
>
> 1 ) Ignoring System 2: We understand this requested change in multiple ways: A) add more high-level effects of structural learning and B) talk about the analogies of system 1 and system 2 in ANNs. We will respond to each below, but would appreciate further clarification on what you are specifically requesting.
>
> A) We understand your point in discussing the higher-level or deeper analysis of the connections between artificial and biological neural networks and agree that this is within scope of the paper. Part of the point of this paper is to point out the diverging disconnect between how ANNs perform structural learning versus how it works in biology: while each ANN structural learning approach should necessarily be optimized for its electronic media, presenting and contrasting them alongside relevant biological methods allows for a more complete picture of how to move forward.
>
> Now that we have integrated the biological section within the sections on pruning and creation and added further analysis here and throughout, we hope this desire is more sufficiently met.
>
> B) We chose to focus on structural learning, going down to the measurable and observable atomic operations, and their physical correlates in biology, even if such structures are what potentiate higher System 2 level effects on behavior. Neuroscientists do not yet fully understand how the low-level cellular processes in the nervous system give rise to high-level capabilities, and we base our arguments in the neuroscience literature. ANNs are also very far from system 2 thinking, although works such as the proposed references are in the right direction. Computers in general are relatively strong at system 1 tasks compared to their biological counterparts. Structural learning algorithms, encoded in computer languages, reflect this: we can easily define a pruning algorithm to optimize an explicitly defined and computed quantity, but we are not yet at the level of telling the model to become more analytical or less biased through structural changes. As the push for ANNs to continue to improve in higher level (i.e. System 2) tasks, structural learning algorithms and techniques may be able to as well. We discuss this push towards higher levels of abstraction in algorithm design in the meta-learning section of 5.2.
>
> 2 ) Missing discussions on generalization:
> -We agree that this is a notable benefit/goal of pruning and have added some discussion and this citation in Section 3.1.
> -Liang et al. 2018 is an interesting theoretical approach to neurogenesis that we were not previously aware of. We have added this citation with discussion twice in Section 4.3.
>
> 3 ) Missed bio-inspired sparse networks: We choose to focus our framework on standard ANNs, using time-independent linear neurons with non-linear activation functions, similar to our decision to exclude spiking neural networks. We believe that this constrained scope allows us to perform deeper analysis about the neural operators when applied to similar neuron and synapse models, such as organization in layers. However, we added related biological networks in Section 5.2.
>
> 4.1.  "Finally, we exclude architectures with self-gating mechanisms, such as LSTMs …"
> These mechanisms have different analogies to the brain, i.e. temporary neural circuitry gating rather than more permanent structural changes to the circuitry; we consider them adjacent to the study case but not within the proposed neural operator framework. We have clarified this paragraph and added further discussion on recurrence and attention as approaches adjacent to structural learning in Section 5.2.
>
> 4.2.  "In biology, the genetic code potentiates each neural operator within biological media…"
> We clarified this statement by summarizing how genes affect structural operations in the biological case, and then qualifying “the components of the algorithm” rather than just “the algorithm” and making the remainder of the sentence more open for the second sentence. We removed our mention of general intelligence from the remainder of the paper and chose to not discuss it here, as it is outside of our scope and expertise to discuss it further.
>
> 4.3. "As computational speed and power continues to increase, the desire to automate the engineering of ANN architectures is increasing …"
> Similarly as above, we removed the latter sentence here.

---

### Author Response · Authors · 2022-06-24
**General initial response to reviews**

We thank each reviewer for their timely reviews on this long paper. We respond to each individually below. New additions are highlighted in red. Regarding the structure of the paper, we removed the section on biology, instead integrating parts of this information throughout the following sections with a stronger link to artificial methods. We also added a subsection on adjacent domains to our final section on perspectives. Thus, the section numbering has changed. All of our responses refer to the current section numbering.

---

### Comment · Reviewer_eh6J · 2022-08-08
**comment re exact copying**

Maybe too late to highlight this, but anyway:

This manuscript contains a verbatim copy of a sentence from a different paper. What is the protocol for this kind of thing - not plagiarism exactly, but verbatim use of other authors' text?

From my original review:
"Page 5:
(*) 4: Are there more references for this? Also, can the Nelson-Alkon ref be moved to connect to this sentence (it's an the exact quote of the first sentence of Nelson-Alkon's abstract)"

---

> ### Author Response · Authors · 2022-08-09
> **Re: "comment re exact copying"**
>
> We appreciate the reviewer's attention to detail in the original review. However, the claimed copied sentence, "...synaptogenesis plays a central role in associative learning and memory," no longer exists in the current published version of the paper. The section that the reviewer was originally referring to has been condensed into "Biological Inspiration" within Section 4.2 on page 16 of the current version, where we no longer cite Nelson & Alkon (2015) nor use similar wording. We apologize for any confusion in previous drafts.

---

### Decision · Action_Editors · 2022-07-14

**Recommendation:** Accept with minor revision

**Comment:**

After a thorough set of reviews and responses from the authors, 2/3 reviewers have recommended acceptance. Based on this, and the authors' responses to the one reviewer who recommended rejection which at least partially address those concerns in my opinion, I believe that this paper passes the bar for acceptance. Please upload a final version of the paper with appropriate formatting (e.g. no coloured text).